# SLEEPY: a comprehensive Python module for simulating relaxation and dynamics in nuclear magnetic resonance

Albert A. Smith ⑥ ✉ & Kai Zumpfe

Nuclear magnetic resonance is a powerful method for characterizing dynamics of biological systems in a native-like environment. Accurate dynamics characterization, however, often requires simulations of complex NMR experiments. While a number of simulation programs exist for NMR simulation (SIMPSON, Spinach, SpinEvolution), none of these are focused on easy simulation of motional effects on NMR experiments. The SLEEPY Python module makes it straightforward to simulate arbitrary pulse sequences while including both relaxation and exchange processes. SLEEPY furthermore allows simulation of solid-state (static and spinning) and solution NMR experiments, using both truncated and full Hamiltonians (rotating frame/lab frame). We demonstrate its application to a wide variety of experiments, including transverse ($T_{1\rho}$), and longitudinal relaxation ($T_1$), nuclear Overhauser effect magnetization transfers, recoupling, and paramagnetic effects. We also provide an extensive online tutorial that explains how to use the various capabilities of SLEEPY. This tool can then be used for both better understanding of the impact of dynamics on NMR and in reproduction of experimental results.

Nuclear magnetic resonance (NMR) has long been a key experimental method for biological structure determination. NMR is uniquely flexible, but highly complex, with nearly endless possibilities for pulse sequences to select different pathways for magnetization, in order to obtain various structural information. Simulations are essential for developing efficient pulse sequences, and for understanding their behavior for different conditions. SIMPSON[1,2], for example, has become very popular for simulating rigid systems, being relatively simple to use, computationally efficient, and free. Less simple, but widely applicable to complex problems is the SPINACH software[3], and powerful but expensive is SpinEvolution[4].

However, a seismic shift in structural biology has been underway for several years now: advances in experimental methods have vastly extended the limits of structure determination. Improved detectors and analysis methods have expanded the applicability of cryo-electron microscopy[5]. X-ray diffraction continues to become more routine, and can be applied to new systems due to the development of X-ray free-electron lasers[6]. Improved cryo-probes, ultra-fast magic-angle spinning[7], and higher magnetic fields increase sensitivity and resolution in NMR[8], allowing studies of increasingly complex systems. The enormous number of protein structures deposited in the RCSB Protein Database (241,055 as of writing)[9] has also allowed the remarkable development of artificial-intelligence based structural models, such as AlphaFold[10], that solve protein structures from amino-acid sequence alone.

These developments create new opportunities for NMR of biological systems. With protein structures available at the click of a button, NMR is more easily extended to the study of *dynamics*. Indeed, biomolecules do not have a single structure, but sample a conformational ensemble, which plays a major role in determining function. Furthermore, proteins interact intimately with their environment, where membrane protein function is modulated by membrane composition, and the resulting dynamics of the membrane[11]. It makes little sense to study motion directly in a frozen or crystalline sample, such that this may be NMR's strongest suit: its ability to investigate motion in a

Institute of Medical Physics and Biophysics, Leipzig University, Leipzig, Germany. ✉e-mail: albert.smith-penzel@medizin.uni-leipzig.de

native-like environment[12,13]. Various experiments in NMR allow the characterization of amplitude and timescale of dynamics via measurement of residual anisotropic couplings, relaxation rate constants, and chemical exchange[14,15]. Other NMR experiments use paramagnetism, where electron relaxation impacts the nuclear signal, in order to improve structural or dynamic characterization[16].

To extract dynamics information, for example, amplitude and correlation times of motion, one must be able to model the data. The NMR community has often relied on analytical formulas based on Bloch-Wangness-Redfield theory[17,18] and the Bloch-McConnell[19] equations. However, to fully capture dynamic effects on a pulse-sequence consisting of more than a simple, continuously applied radiofrequency field (such as the $T_{1\rho}$ experiment[20]), often we require a simulation[21,22]. Then, to support development of new NMR methods for dynamics characterization and to promote a better understanding of the impact of relaxation and dynamics on NMR spectra, simulation software is required for the simulation of NMR experiments under these conditions.

For this purpose, we developed the *Spins in Liouville-space for rElaxation and Exchange in PYthon* (SLEEPY) simulation package. SLEEPY was developed to make it as simple to simulate a pulse sequence with dynamics/relaxation as it is to simulate the same sequence without these effects (e.g., via SIMPSON[1,2]). We furthermore wanted an interactive program that would provide useful information while setting up a simulation and allow access to internal components of the simulation to obtain insights into how a simulation worked. Quick sharing is also critical, to provide both an educational tool and better reproducibility. Simulations should not be a black-box that only computational experts can verify, it should be possible for anyone to quickly re-run a simulation and verify that parameters impact results as expected; given today's availability of online computational resources, this can easily be done in a web-browser (furthermore, it can be done cost-free). This has been achieved with SLEEPY's object-oriented implementation in Python 3.

## Results and discussion
### Simulation basics
A simulation in magnetic resonance has the basic task of evaluating how a spin system with a certain set of interactions evolves in time, and monitors the evolution of one or more terms describing the state of the spin system. Then, the state of the spin-system is described by a density matrix, $\hat{\rho}(t)$, the evolution is brought about by a Hamiltonian ($\hat{H}$, Hilbert space) or Liouvillian ($\hat{\hat{L}}$, Liouville space), and evolution is monitored with detection operators ($\hat{O}$). SLEEPY simulations are performed in Liouville space, which allows one to propagate the spin-system both due to coherent effects described by the Hamiltonian, but also from exchange processes which result in stochastic changes to the Hamiltonian, and relaxation processes, which bring about non-coherent transfer of magnetization between states, or destruction of magnetization. In Liouville space, the current state of the system is stored as vector, which gives the expectation values for the basis set (SLEEPY uses the Hilbert basis, so if we have a two-spin system, these include $\langle \hat{S}_1^\alpha \hat{S}_2^\alpha \rangle(t)$, $\langle \hat{S}_1^\alpha \hat{S}_2^+ \rangle(t)$, $\langle \hat{S}_1^\alpha \hat{S}_2^- \rangle(t)$, $\langle \hat{S}_1^\alpha \hat{S}_2^\beta \rangle(t)$, $\langle \hat{S}_1^+ \hat{S}_2^\alpha \rangle(t)$, etc.). Propagators bring $\hat{\rho}(t)$ forward in time, via matrix multiplication, e.g., $\hat{\rho}(t_1) = \hat{U}(t_0, t_1)\hat{\rho}(t_0)$. Propagators can be calculated from a time-independent Liouvillian, but may also account for time-dependence, arising from both rotor spinning and pulse sequences (Liouvillian dependence on the rotor position results from changes in the Hamiltonian due to reoriented anisotropic terms, but may also result from orientation-dependent relaxation behavior). In SLEEPY, time dependence is handled by approximating the continuously varying Liouvillian as a discrete

series of time-independent Liouvillians, resulting in, for example, $\hat{U}(t_0, t_n) = e^{\hat{\hat{L}}(t_{n-1})(t_n - t_{n-1})} \cdot \ldots \cdot e^{\hat{\hat{L}}(t_1)(t_2 - t_1)} \cdot e^{\hat{\hat{L}}(t_0)(t_1 - t_0)}$, where the individual Liouvillians, $\hat{\hat{L}}(t_n)$, are assumed to be time-independent.

SLEEPY returns the expectation value of a user-specified detection operator or operators whenever a detection operation is executed. This calculates the dot product $\langle \hat{O} \rangle(t) = \hat{O} \cdot \hat{\rho}(t)$, where $\hat{O}$ is the detection operator. SLEEPY allows setting the detection operator via text-input for specific spins or specific nuclei (e.g., $\hat{S}_{0x}$: "`S0x`", $\hat{S}_{2y}$: "`S2y`", $\sum_{i \in {}^1H} \hat{S}_i^+$: "`1Hp`", etc.) or via user-provided matrices for more complex operators, and supports multiple detection operators. Note that both propagation and detection are performed over all elements of a powder average, if anisotropic interactions are included in the simulation.

The Liouvillian in SLEEPY is constructed from coherent terms arising from the Hamiltonian (or Hamiltonians in exchange) including the time-dependent radiofrequency field, and additionally from non-coherent exchange terms and relaxation terms. The coherent terms in the Liouvillian may be constructed with all spins in the rotating frame (secular approximation), all spins in the lab frame, or in a mixed frame (pseudosecular approximation). Application of the rotating frame removes terms from the Hamiltonian that rotate in the interaction frame of the Zeeman interaction, and removes the Zeeman interaction itself. This can significantly accelerate simulations, but in this case simulations will not produce certain exchange-induced relaxation processes such as $T_1$ and heteronuclear NOE[23] (see "Methods" for details).

Exchange in SLEEPY is introduced by defining two or more total Hamiltonians, where one or more interactions in the Hamiltonians have either varying magnitude or different orientations among the Hamiltonians. SLEEPY then expands each into a Liouvillian and couples them together via an exchange matrix. Relaxation in SLEEPY is added by introducing decay on terms in $\hat{\rho}_\Omega(t)$ (e.g., $T_2$ decay), or by added exchange between terms in $\hat{\rho}_\Omega(t)$ to include $T_1$ processes. Relaxation is implemented along the cartesian coordinates ($T_1$ along $z$, $T_2$ in the $xy$-plane), or alternatively in the eigenbasis ($T_1$ between eigenstates, $T_2$ on coherences). Thermalization of the spin-system[24–26], i.e., recovery of the system to thermal equilibrium, may be included when $T_1$ is added via Lindblad thermalization[27], or may be added to thermalize systems with exchange-induced relaxation (non-Lindblad approach[26]).

SLEEPY includes a number of features to accelerate calculations. This includes optional parallel execution of the powder average, propagator caching[28], eigenbasis propagation[29], and exact basis-set reduction (approximated basis set reduction according to spin-order is not implemented, where approximate reduction enables simulating larger spin-systems as is done in SPINACH[3,30]). SLEEPY has been benchmarked for simulations with up to 6 spins, where calculations in Liouville space typically increase in computational time by a factor of about ~50 for each additional spin-1/2 (scaling ~$4^{2.81}$)[31].

While we have provided a brief explanation of SLEEPY functionality here, many more details can be found in the "Methods" section.

### Easy, interactive setup
Figure 1A shows the steps for calculating relaxation under a spin-lock, where $^{13}C$ $T_{1\rho}$ relaxation (transverse relaxation) is induced by a stochastic 15° hop of a heteronuclear dipole coupling between a $^1H$ and $^{13}C$ nucleus, with a 1 μs correlation time. $T_{1\rho}$ is a critical experiment for characterizing the correlation time of slow motion in the range of milliseconds to high nanoseconds, where one varies the applied field strength of the spin-lock in order to be able to fit correlation times and amplitudes of motion to the relaxation rates[32–34]. The first step to simulate $T_{1\rho}$ is to build two experimental systems ("`ExpSys`" objects "`ex0`" and "`ex1`"). This is how we introduce motion, where the two

## A   Full simulation code

```
ex0=ExpSys(v0H=600,Nucs=['1H','13C'],vr=45000)    # Field (MHz), Nuclei, MAS frequency
ex0.set_inter('dipole',i0=0,i1=1,D=22000)         # Dipole, spins 0,1, D=22 kHz
ex1=ex0.copy()                                     # Second system
ex1.set_inter('dipole',i0=0,i1=1,D=22000,euler_d=[0,15,0])# Same dipole, with 15 deg β-angle
L=Liouvillian(ex0,ex1,kex=Tools.twoSite_kex(tc=1e-6))  # Couple systems in Liouvillian
seq=L.Sequence().add_channel('13C',v1=35000)       # Pulse sequence, 35 kHz 13C field
rho=Rho(rho0='13Cx',detect='13Cp')                 # Density matrix/detection
rho.DetProp(seq,n=4000)                            # 4000 steps (88.9 ms)
```

## B   Text output examples

| ex0 |
| --- |

```
2-spin system (1H,13C)
B0 = 14.092 T (600.000 MHz 1H frequency)
rotor angle = 54.736 degrees
rotor frequency = 45.0 kHz
Temperature = 298 K
Powder Average: JCP59 with 99 angles
Interactions:
        dipole between spins 0,1 with arguments:
                (D=22000.00)
```

| rho |
| --- |

```
Density Matrix/Detection Operator
rho0: 13Cx
detect[0]: 13Cp
Current time is 88888.889 microseconds
4000 time points have been recorded
```

## C   Graphical output examples

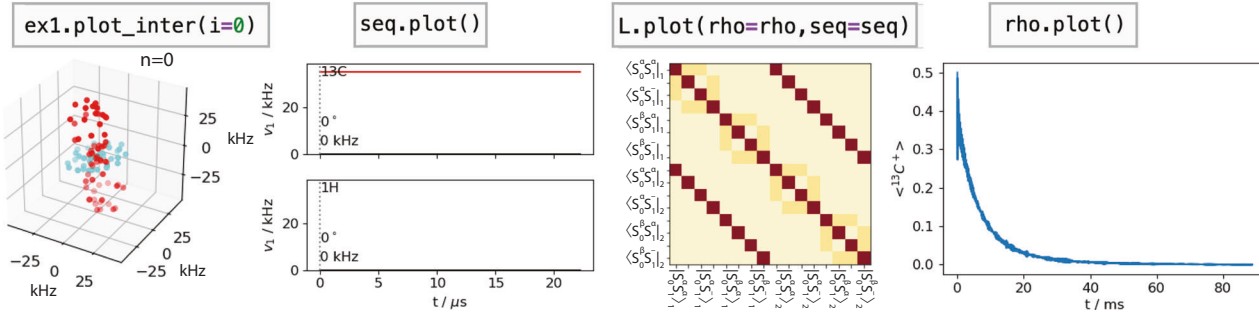

**Fig. 1 | Code and outputs for simulation of $^{13}C$ $R_{1\rho}$ induced by a 15° reorientation of an $^{1}H$–$^{13}C$ dipole coupling. A** shows the full simulation code, starting with defining the experimental conditions and adding interactions (ex0,ex1), followed by defining the Liouvillian with exchange with $\tau_c = 1\,\mu s$, and adding a pulse sequence with a field applied on the $^{13}C$ channel. Finally, a density matrix with an initial orientation along the x-direction on $^{13}C$ is generated, with detection on $^{13}C^+$, and is propagated for 4000 repetitions of the sequence. **B** shows the text-based description of two objects for example, and **C** shows plotting functions for several of the objects in SLEEPY. All user-accessed objects in SLEEPY provide similar text-based description of themselves, and also include a plotting function to display the data in the object. Source data for C are provided as a Source Data file.

systems are constructed with the same interactions, but with a change in the orientation of the dipole coupling. We could easily add additional interactions, for example, chemical shift and chemical shift anisotropy both might also be modulated by motion. A Liouvillian is constructed from "ex0" and "ex1" with a matrix describing the exchange between all "ExpSys" objects ("kex = …"). A sequence ("seq") can be created from the Liouvillian, and allows both simple, continuous radio-frequency fields, but also complex, multi-channel sequences with varying field strength, phase, and offset. Finally, a density matrix ("rho") defines the initial state of the system and one or more matrices to detect.

The whole simulation requires eight lines of code (a few of which could be consolidated). A simulation can be run a few lines at a time, while checking the status of each created object via text and graphical outputs to ensure that all parameters are as expected. Figure 1B shows text outputs from two of the objects, and Fig. 1C shows the graphical output from the objects. All SLEEPY objects produce similar text and graphical output, in order to provide the user feedback during setup and give insight into the simulation components and results.

### Simulating complex sequences

The $T_{1\rho}$ experiment is a very simple pulse sequence, but REDOR[35] and DIPSHIFT[36], and sophisticated methods to measure $T_{1\rho}$, such as the RECRR experiment[22] require execution of more involved pulse sequences[37]. SLEEPY's implementation of pulse-sequences and propagators makes this straightforward. We use the RECRR $T_{1\rho}$

measurement as an example how to build such a sequence, with the desired sequence shown in Fig. 2A. The sequence sandwiches two refocusing periods (refA, refB) each between 2(n–1) rotor periods with a spin-lock on x or -x (cwx, cwmx). The relaxation decay curve is obtained by stepping n from 1 up to N rotor periods (for a total spin-lock length of 4n rotor periods). Definition of spin-locking periods (cwx, cwmx) and refocusing periods (refA, refB) are demonstrated in Fig. 2B. The sequence is then executed by calculating propagators (cwx:Ux, cwmx:Umx, refA:UA, refB:UB) for each period, and at every step increasing the UA and UB propapator lengths by multiplying by Ux or Umx on both sides, in order to increase the total length of the spin-lock. This is shown in Fig. 2D. While some planning is required to efficiently execute sequences, the handling of sequences and propagators is straightforward, where a new propagator may be obtained simply from the product of two or more other propagators. Furthermore, propagators and sequences may be multiplied together, and both may be multiplied by density matrices. SLEEPY will correctly handle timing for each operation if a sequence is involved, and provides warnings in case the user mis-times application of a propagator.

### A broad range of applications made simple

SLEEPY may be applied to a broad range of problems involving relaxation and/or exchange processes in NMR. SLEEPY features the ability to perform solution, static, and spinning (by default, magic-angle spinning, MAS) simulations in either the rotating or lab-frame. It includes simple implementation of exchange, addition of $T_1$ and $T_2$

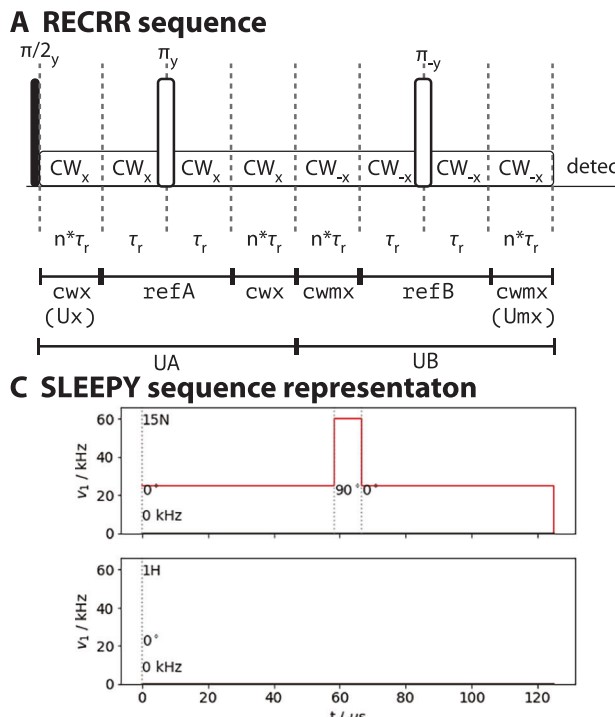

**A  RECRR sequence**

**B  Sequence setup**

```
v1=25000
v1pi=60000
pi2=1/v1pi/2

cwx=L.Sequence()
cwx.add_channel('15N',v1=v1)     #Spin-lock on x
cwmx=L.Sequence()
cwmx.add_channel('15N',v1=v1,phase=pi) #Spin-lock on –x

t=[0,L.taur-pi2/2,L.taur+pi2/2,2*L.taur]
#First refocusing period
refA=L.Sequence()
refA.add_channel('15N',t=t,v1=[v1,v1pi,v1],phase=[0,pi/2,0])
#Second refocusing period
refB=L.Sequence()
refB.add_channel('15N',t=t,v1=[v1,v1pi,v1],phase=[pi,3*pi/2,pi])
```

**C  SLEEPY sequence representaton**

**D  Execute sequence**

```
rho=sl.Rho(rho0='15Nx',detect='15Nx') #Density matrix

Ux=cwx.U()    #Propagator for x spin-lock
Umx=cwmx.U()  #Propagator for –x spin-lock
UA=refA.U()   #First half of sequence
UB=refB.U()   #Second half of sequence

rho()
for n in range(200):
      rho.reset()  #Reset density matrix
      UB*UA*rho    #Propagate by UA and UB
      rho()        #Detect
      UA=Ux*UA*Ux    #Add 1 rotor period to each side of UA (x)
      UB=Umx*UB*Umx  #Add 1 rotor period to each side of UB (–x)
```

**Fig. 2 | Simulating a complex pulse sequence in SLEEPY. A** shows the desired pulse sequence for the REfocused CSA Rotating-frame Relaxation (RECRR) experiment. Below the sequence is shown how to efficiently break the full sequence into parts for simulation. Then, `cwx` and `cwmx` define the continuous wave spin-lock without pulses for one rotor period, and `refA` and `refB` define the refocusing periods (two rotor periods each). **B** shows how to code each part of the sequence. Sequences are by default one rotor period, so `cwx` and `cwmx` do not require explicit definition of the time axis, but the refocusing periods (`refA`, `refB`) require explicit definition of their time axis to obtain the correct length of π-pulse. **C** shows the

SLEEPY representation of the `refA` sequence, where we see the field strength plotted as a function of time for both channels. Phases (0°, 90°, 0°) and offsets (0 kHz) are printed on the plot. **D** shows execution of the sequence, which starts with propagators `UA` and `UB`, generated from `refA` and `refB`. Then at every time point, one rotor period of continuous irradiation is added to either side of `UA` and `UB`. Note that at every element of the "for" loop, we start again at $t = 0$ s, achieved by calling `rho.reset()`. Then, `rho` is propagated with `UB` and `UA` (`UB*UA*rho`), followed by signal detection (`rho()`). After this, `UA` and `UB` are extended (`UA = Ux*UA*Ux`, `UB = Umx*UB*Umx`) for the next time point.

either along the *z*-axis and *xy*-plane, or in tilted frames (i.e., enforcing $T_1$ relaxation between eigenstates of the Hamiltonian and $T_2$ relaxation along coherences between those eigenstates), and Lindblad thermalization to return populations to their thermal equilibrium values, while also obtaining correct relaxation of the coherences[27]. A "dynamic" thermalization is available, to recover populations that decay due to exchange-induced relaxation. Arbitrary pulse sequences may be specified for each nucleus type or for the individual spins. Figure 3 shows a sample of applications of SLEEPY, where different features of the software are applied. Of the examples, all but H and I involve some type of explicit simulation of exchange, with chemical shift modulation in A, B, and E, and reorientational dynamics used in C, D, E, F, G (G also includes modulation of the isotropic hyperfine coupling). Arbitrary pulse-sequence generation is utilized in A, B, and D. Simulations with longitudinal relaxation induced by motion and the pseudocontact shift (PCS) must be performed in the lab frame, which includes C, F, G, H, and I. Specialized tools are included in SLEEPY and applied in A, for constructing a 2D experiment, in C and H for constructing exchange to mimic tumbling, in F for extracting individual rate constants from the powder average, and in G for applying a saturating microwave field to the electron while simulating in the lab frame.

The PCS represents a particularly challenging case for simulation. PCS (Fig. 3H, I) is induced by a fast-relaxing electron with unequal spin-up ($\hat{S}^\alpha$) and spin-down ($\hat{S}^\beta$) populations, such that the electron-nuclear dipole splitting is removed but the electron polarization leaves one manifold more populated than the other, i.e., $\left\langle \hat{S}^\alpha \hat{I}^+ \right\rangle < \left\langle \hat{S}^\beta \hat{I}^+ \right\rangle$. Then, a residual shift manifests on the nucleus where the splitting does not average to zero. In Fig. 3H, PCS is simulated under solution conditions,

where the relative orientations of electron-nuclear hyperfine couplings and electronic g-tensor are varied, in order to produce the shape of the PCS angular dependence. In Fig. 3I, the PCS lineshape under MAS is shown, demonstrating that PCS is not fully removed by MAS. Key points in simulating PCS are highlighted in Fig. 4A. Simulations must be performed in the lab frame, to obtain the correct tilting of the electron due to its *g*-tensor anisotropy (Fig. 4A, point i). For solution-state PCS, we require dynamics that mimic tumbling, achieved here with a tumbling model with 10 sets of Euler angles distributed around a sphere (ii) (see "Methods" for an explanation of the tumbling models and proper usage). Furthermore, PCS requires correct thermalization of coherences (Lindblad thermalization[27]), otherwise populations of the $\hat{S}^\alpha \hat{I}^+$ and $\hat{S}^\beta \hat{I}^-$ coherences will rapidly equilibrate and destroy the contact shift (iii). Finally, PCS requires electron relaxation to be implemented in a tilted frame, such that applied relaxation rate constants are not mixed between electron and nucleus and between $T_1$ and $T_2$, and polarization must be adjusted as a function of orientation (iv). Points v and vi highlight more basic settings: temperature settings (v) and execution of the sequence and plotting (vi). While there are a few settings to keep track of for a more complicated simulation such as PCS, all are easily included in the simulation with just a few lines of code. Figure 4B shows the solution-state PCS as a function of temperature, where decreasing temperature increases the total nuclear signal and increases in electron polarization increase the magnitude of the shift.

**Complexities of simulation**

While we have made an effort to make setup in SLEEPY fairly simple, one does need to keep in mind that ultimately, the best simulation

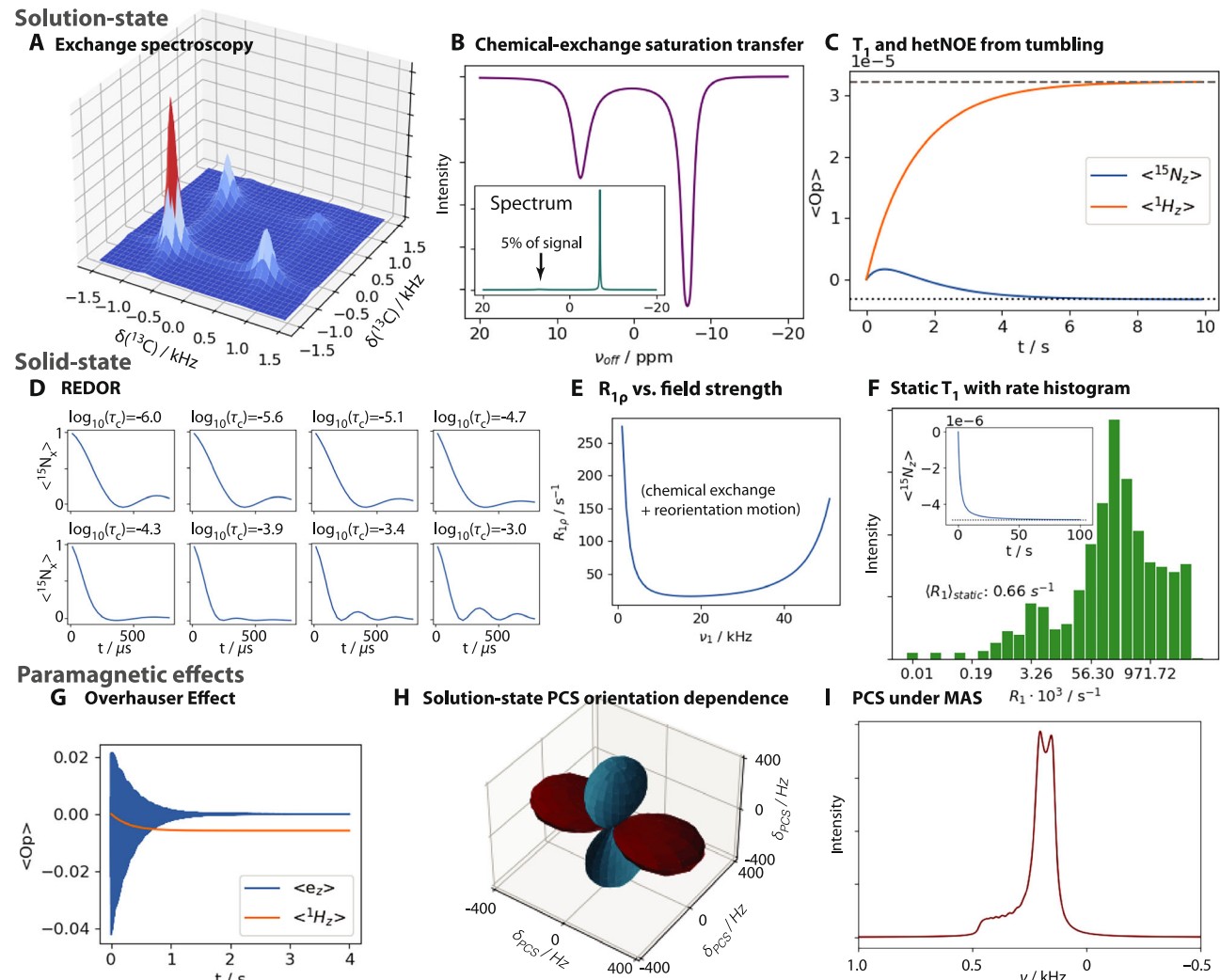

**Fig. 3 | Diverse applications of SLEEPY. A** Typical exchange spectroscopy (EXSY) experiment, with two chemical shifts in asymmetric exchange (1-spin $^{13}$C system). **B** Saturation profile obtained for a chemical exchange saturation transfer (CEST) experiment (1-spin $^{13}$C system). **C** $T_1$ and heteronuclear nuclear Overhauser effect (NOE) affects induced by a simple model of isotropic tumbling in solution (2-spin $^{15}$N, $^1$H system). Dotted/dashed lines indicate thermal polarization of the spins. **D** Rotational-Echo DOuble-Resonance (REDOR) dephasing curves as a function of correlation time (2-spin $^{15}$N, $^1$H system). **E** $^{15}$N $R_{1\rho}$ relaxation rate constant as a function of field strength (60 kHz magic-angle spinning, MAS) for a system with both chemical exchange and reorientational dynamics (2-spin $^{15}$N, $^1$H system).

**F** Anisotropic $T_1$ relaxation for a static sample, where all relaxation rate constants are extracted and plotted in a histogram. The inset gives the original $T_1$ decay curve, with thermal equilibrium marked with a dotted line (2-spin $^{15}$N, $^1$H system). **G** Overhauser effect DNP enhancement of $^1$H signal via electron microwave irradiation (2-spin e-, $^1$H system). **H** Orientation dependence of the pseudocontact shift (PCS) in solution-NMR (2-spin e-, $^1$H system). The distance of the surface from the origin indicates the size of the shift, where a maroon surfaces indicate a positive shift, whereas turquoise indicates a negative shift. **I** Pseudocontact shift lineshape in solid-state NMR under MAS (2-spin e-, $^1$H system. Source data are provided as a Source Data file.

software can do is correctly simulate the system under the conditions provided by the user. This means that the user needs to be aware what they have simulated and if it really makes sense for the problem at hand. SLEEPY provides a number of tools for investigating the simulation, e.g., plotting functions of the powder average, interactions, and matrices, and descriptive objects (Fig. 1B, C). All matrices used for simulation are also user-accessible. Then, the user needs to take care to ensure that parameters are properly set. For example, by default, SLEEPY uses a relatively small powder average, but for certain applications (e.g., REDOR), this will be insufficient. Ultimately, the user needs to experiment with the various settings (powder average type, number of angles, number of steps, i.e., γ-angles, per rotor period), to ensure that simulations produced are sufficiently accurate for the desired application. SLEEPY indeed provides quite a few default settings, with the goal of getting users off to a quick start, but this can also be dangerous; users still need to be aware that for accurate simulations, these settings may still need

to be adjusted. We encourage the users to check their simulation setups using the text- and graphical outputs highlighted in Fig. 1B, C, and check their results by adjusting the simulation parameters and verifying stability, e.g., as a function of powder average size, type, and number of γ-angles.

SLEEPY includes a number of methods for including $T_1$ and $T_2$ relaxation. While new methods can be added without restructuring the core code, currently $T_1$ and $T_2$ relaxation is added only with a fixed value for all orientations in the powder average. This phenomenological approach allows one to determine the effect of relaxation of one part of the system on another (e.g., PCS in Fig. 3H, I, Fig. 4), but it may not be appropriate for models of some real systems, where relaxation is multi-exponential due to orientation dependence (e.g., Fig. 3F, where the multiexponential relaxation not introduced directly, but is an indirect effect of the exchange processes).

In the simplest case, $T_1$ relaxation can be set to act simply along the $z$-axis, $T_2$ affects relaxation in the $xy$-plane, and the thermal

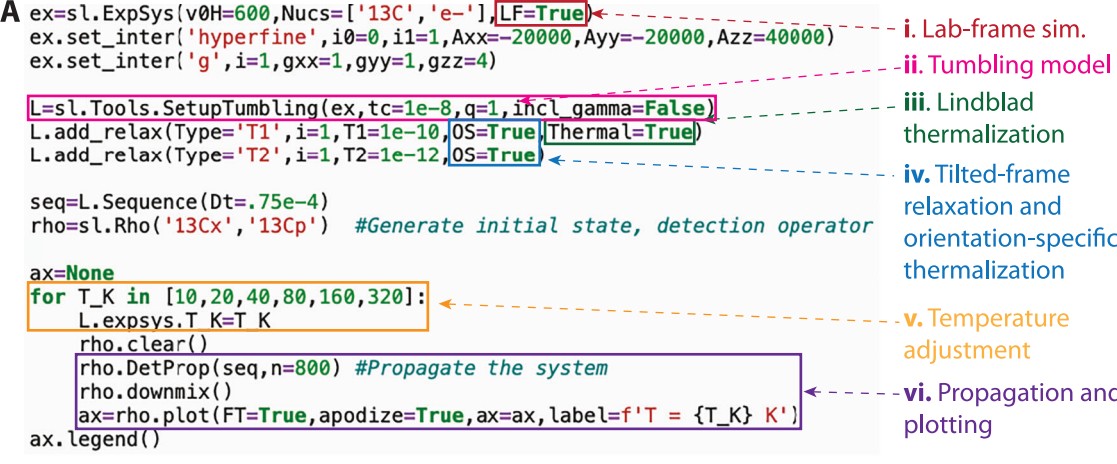

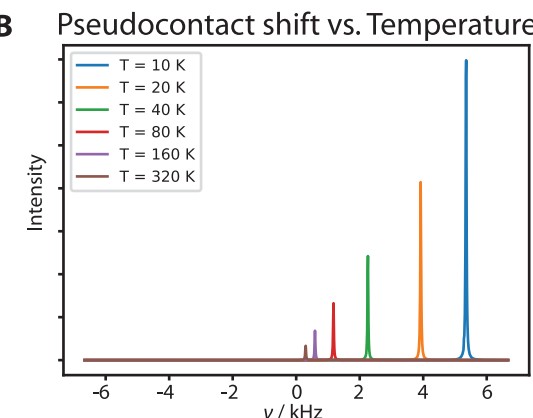

**Fig. 4 | Setup of a pseudocontact shift simulation in solution state NMR.**
**A** highlights the key points required for setup, which are marked. i: Lab-frame simulation, ii: Tumbling model in solution, iii: Lindblad thermalization, iv: tilted-frame relaxation with orientation-specific thermalization. v and vi are more standard settings, highlighting where the temperature gets changed (v) and where the experiment is propagated and plotted (vi, including downmixing from the lab-frame to the rotating frame). **B** shows the shift in resonance frequency resulting from the pseudocontact shift as a function of temperature, where decreasing temperature increases the nuclear signal, and also increases the size of the shift due to increasing electron polarization. Source data for B is provided as a Source Data file.

equilibrium is fixed to an average value (SLEEPY's default behavior). However, SLEEPY also gives the option to provide relaxation in the eigenbasis, which means $T_1$ relaxation is always an equilibration of eigenstates and $T_2$ relaxation always destroys coherences (used in Fig. 4A, set with "$OS = True$"). This mode furthermore adjusts thermal equilibrium throughout the rotor period. Eigenbasis relaxation can be important in systems where the quantization axis of a spin is tilted away from the z-axis, or if spin states are mixed. Eigenbasis relaxation must be used, for example, for PCS, where tilting of the electron away from the z-axis is important for the effect. On the other hand, eigenbasis relaxation will not induce some types of paramagnetic relaxation enhancement originating from the electron $T_2$ relaxation, since it will remove mixing of the electronic and nuclear states via non-secular terms in the Hamiltonian. Then, for some effects, especially processes occurring in tilted frames, one needs to be very aware of the significance of the type of relaxation applied and the limitations thereof.

## Learning by simulation and sharing results
An important goal of the SLEEPY software is to provide an educational tool to help the NMR community better understand dynamic effects and their impact on NMR experiments. Then, along with the SLEEPY software, we release the SLEEPY tutorial, available at http://sleepy-nmr. org/. This tutorial provides explanations of the various SLEEPY objects and starts with basic examples, followed by sections covering 21 different types of NMR experiments. Each simulation can be opened from a link on the webpage in one of two cloud-computing servers, Google

Colab and MyBinder, so that one can immediately execute the simulations in a browser without any local installation. Users can also share their simulations, where simply uploading a Jupyter Notebook to a public GitHub repository allows other users to view and edit simulations with the Github-to-Colab functionality (see "Sharing SLEEPY Simulations" in the SLEEPY tutorial for details). Note the recently-published software MRSimulator, which has also been developed in Python for simulating solid-state NMR lineshapes, can also be shared this way[38]. We will also host simulations on the SLEEPY website from any researchers wanting to share their own code.

Other programs exist for some of the simulations demonstrated here in SLEEPY, in particular Spinach[3], GAMMA[39], and SpinEvolution[4] allow the addition of various relaxation or exchange processes. However, to the best of our knowledge, exchange-induced longitudinal relaxation ($T_1$, Overhauser and nuclear Overhauser effects) and PCS have not been implemented in these programs. Even measurement of residual dipolar tensors under exchange conditions (e.g., via REDOR) has only been recently implemented in GAMMA[21], and does not appear in the Spinach and SpinEvolution libraries. Furthermore, the possibility of implementation an experiment in a given program is not the only question to consider, since in principle, most programming languages allow arbitrary NMR simulations (e.g., we have used MATLAB for $R_{1\rho}$ simulations without any NMR simulation library[33]). The more relevant question is how much knowledge and effort are required by the user to generate those simulations. This is where SLEEPY excels: by already being set up to simultaneously include exchange, relaxation, and

thermalization for a variety of experiments and pulse-sequences in NMR, the user is not required to have significant background knowledge in NMR computation. SLEEPY also has the advantage that it is fully cost-free, whereas SpinEvolution charges a yearly fee, and Spinach requires a MATLAB subscription. Finally, SLEEPY is easy to share: one click in the SLEEPY tutorial brings the user to a ready-to-run Python environment, whereas no such share option is available for GAMMA or SpinEvolution. Matlab Online allows running Spinach in a browser, but there is no pre-setup environment or simulations currently available online.

With changes in the landscape of the field of structural biology, many expert NMR groups may increasingly focus on characterizing dynamics, where NMR has a clear advantage over other methods such as X-ray, cryo-EM, and AI-based approaches. However, groups whose main focus is not NMR simulation can benefit significantly from simple-to-use software for simulating various dynamic effects. We believe that SLEEPY and its tutorial can provide such a tool to educate users on dynamics effects, to reproduce experiments, and to simplify the sharing of simulations, thus helping to expand the application of NMR to characterization of complex dynamics processes.

## Methods
### Propagation in Liouville space
All propagation steps in SLEEPY are performed in Liouville space, such that given a propagator, $\hat{U}_\Omega(t_1 t_0)$ (propagator for Euler angles, $\Omega$, which brings the system from time $t_0$ to time $t_1$), and density matrix, $\hat{\rho}_\Omega(t_0)$, we may calculate $\hat{\rho}_\Omega(t_1)$ according to

$$\hat{\rho}_\Omega(t_1) = \hat{U}_\Omega(t_0, t_1) \cdot \hat{\rho}_\Omega(t_0) \tag{1}$$

The density matrix is a column vector (and the propagator a matrix) whose initial value is obtained by collecting the columns of the desired square density matrix (e.g., $\hat{S}_{1x}$, $\hat{S}_{2z}$) one after the other in a single column. For brevity, we will drop the Euler angle subscript from here, but keep in mind that these calculations may be performed for different elements of a powder average.

To obtain the propagator for a static Hamiltonian, we first calculate the corresponding Liouvillian:

$$\hat{L}(t_0) = -i\left(\hat{H}(t_0) \otimes \hat{1} - \hat{1} \otimes \hat{H}^T(t_0)\right) \tag{2}$$

Here, $\otimes$ is the Kronecker product, $\hat{1}$ is an identity matrix the same size as the Hamiltonian, and $\hat{H}^T(t_0)$ is the transpose of the Hamiltonian (not the complex transpose!). For an $n \times n$ Hamiltonian, this yields an $n^2 \times n^2$ Liouvillian. The propagator for a static Liouvillian is then obtained by calculating

$$\hat{U}(t_0, t_1) = \exp\left[\hat{L}(t_0)(t_1 - t_0)\right] \tag{3}$$

For spinning experiments, we may calculate the time dependence of the Hamiltonian introduced by spinning according to

$$\hat{H}(t) = \sum_{n=-2}^{2} e^{-in\omega_r t} \hat{H}_n \tag{4}$$

The $\hat{H}_n$ are the sum over the $n$th rotating components of the spinning of all interactions in the system (chemical shift, J-coupling, dipoles, chemical shift anisotropy, quadrupole, etc.), and $\omega_r$ is the spinning frequency (rad/s). Isotropic interactions only contribute to the non-rotating component, $n = 0$, whereas anisotropic interactions

contribute to all terms, calculated from the spherical interaction tensors. The sum over all interactions is as follows:

$$\hat{H}_n = \sum_i \sum_{l=0,2} \sum_{q=-l}^{l} (-1)^q d_{n,q}^{(l)}(\theta_r) A_{l,n}^{(i)} \hat{T}_{l,-q}^{(i)} \tag{5}$$

The outer sum is over all interactions, $i$, the next sum is over tensor ranks $l = 0$ and $2$, and the inner sum is over components $q$ running from $-l$ to $l$. Then, the $A_{l,n}^{(i)}$ give the interactions in the rotor frame for the given rank and component. $d_{n,q}^{(l)}(\theta_r)$ is the rank-$l$ Wigner rotation matrix, and $\theta_r$ the rotor angle (for rank-0, this is one). $\hat{T}_{l,-q}^{(i)}$ is the corresponding spherical tensor. Rotating terms ($|q| > 0$) of $\hat{T}_{l,-q}^{(i)}$ are dropped when performing rotating frame calculations. If all spins are in the rotating frame, then only the terms $\hat{T}_{2,0}^{(i)}$ and $\hat{T}_{0,0}^{(i)}$ remain, and the form of $\hat{T}_{2,0}^{(i)}$ is modified depending, for example, if it is applied to a heteronuclear or homonuclear spin-pair. In the lab frame, all terms are kept, and in mixed frames, only some of the terms remain (see "Lab and rotating frames").

In SLEEPY, we construct rotating components of the Liouvillian from rotating components of the Hamiltonian, and construct the coherent part of the full Liouvillian from those components.

$$\hat{L}_n = i(\hat{H}_n \otimes \hat{1} - \hat{1} \otimes \hat{H}_n^T)$$

$$\hat{L}(t) = \sum_{n=-2}^{n} e^{-in\omega_r t} \hat{L}_n \tag{6}$$

Then, time dependence resulting from rotor spinning and radio-frequency pulse switching is achieved simply by breaking the evolution into $n$ steps with static Hamiltonians,

$$\hat{U}\left(t_i, t_f\right) = \hat{U}(t_{n-1}, t_n) \cdot \hat{U}(t_{n-2}, t_{n-1}) \cdot \ldots \cdot \hat{U}(t_1, t_2) \cdot \hat{U}(t_0, t_1) \tag{7}$$

where each of the individual propagators is calculated from a static Hamiltonian. SLEEPY will break the rotor period up into "n_gamma" steps, and will additionally break up the propagators where the applied fields' amplitudes, phases, or offsets change.

### Detection
If we want to detect a given operator, $\hat{O}$, we must first take its complex transpose, followed by transforming the operator from Hilbert space to Liouville space (achieved by simply gathering the columns into one extended column). In Liouville space, we may simply calculate

$$\langle \hat{O} \rangle(t) = \hat{O}^\dagger \cdot \hat{\rho}(t) \tag{8}$$

Then, $\cdot$ indicates the dot product of the two vectors. Below, we will also note that this detection may be performed in the eigenbasis of a propagator (see "Eigenbasis propagation").

### Orientation dependence of anisotropic interactions and powder averaging
For a given anisotropic interaction, we have components in the principal axis system (PAS) of the interaction, given by

$$A_{2,0}^{PAS} = \sqrt{\frac{3}{2}}\delta, A_{2,\pm1}^{PAS} = 0, A_{2,\pm2}^{PAS} = -\frac{1}{2}\delta\eta \tag{9}$$

**Table 1 | Rank-2 Spherical tensors depending on choice of lab or rotating frame**

| | $\hat{T}_{2,0}$ | $\hat{T}_{2,\pm1}$ | $\hat{T}_{2,\pm2}$ |
|---|---|---|---|
| Spin-Field (LF) | $\sqrt{\frac{2}{3}}\hat{S}_z$ | $\frac{1}{2}\hat{S}^{\pm}$ | 0 |
| Spin-Field (RF) | $\sqrt{\frac{2}{3}}\hat{S}_z$ | 0 | 0 |
| Spin-Spin (LF/LF) | $\frac{1}{\sqrt{6}}\left[3\hat{S}_{1z}\hat{S}_{2z}-\vec{S}_1\cdot\vec{S}_2\right]$ | $\mp\frac{1}{2}\left[\hat{S}_1^{\pm}\hat{S}_{2z}+\hat{S}_{1z}\hat{S}_2^{\pm}\right]$ | $\frac{1}{2}\hat{S}_1^{\pm}\hat{S}_2^{\pm}$ |
| Spin-Spin (LF/RF) | $\frac{2}{\sqrt{6}}\hat{S}_{1z}\hat{S}_{2z}$ | $\mp\frac{1}{2}\hat{S}_1^{\pm}\hat{S}_{2z}$ | 0 |
| Spin-Spin (RF/LF) | $\frac{2}{\sqrt{6}}\hat{S}_{1z}\hat{S}_{2z}$ | $\mp\frac{1}{2}\hat{S}_{1z}\hat{S}_2^{\pm}$ | 0 |
| Spin-Spin (RF/RF, homonuclear) | $\frac{1}{\sqrt{6}}\left[3\hat{S}_{1z}\hat{S}_{2z}-\vec{S}_1\cdot\vec{S}_2\right]$ | 0 | 0 |
| Spin-Spin (RF/RF, heteronuclear) | $\frac{2}{\sqrt{6}}\hat{S}_{1z}\hat{S}_{2z}$ | 0 | 0 |

$\hat{T}_{2,n}$ are the rank-2 spherical tensors for spin, whereas $\hat{S}_{k,z}$, $\hat{S}_k^{\pm}=\frac{1}{2}\left(\hat{S}_{k,x}\pm i\hat{S}_{k,y}\right)$ are the spin matrices for single spins.
RF and LF refer to the rotating frame and lab frame, respectively.

These are first brought into the molecular frame, using the Euler angles $\Omega^{PAS\rightarrow MF}=\left[\alpha^{PAS\rightarrow MF},\beta^{PAS\rightarrow MF},\gamma^{PAS\rightarrow MF}\right]$ according to

$$A_{2,q}^{MF}=\sum_{q'=-2}^{2}D_{q',q}^{(2)}\left(\Omega^{PAS\rightarrow MF}\right)A_{2,q'}^{PAS}$$

$$=\sum_{q'=-2}^{2}e^{-i\alpha^{PAS\rightarrow MF}q'}d_{q',q}^{(2)}\left(\beta^{PAS\rightarrow MF}\right)e^{-i\gamma^{PAS\rightarrow MF}q}A_{2,q'}^{PAS} \qquad (10)$$

Note that SLEEPY allows entry of multiple sets of Euler angles for an interaction, which is sometimes convenient for defining spin-system geometry, or for introducing exchange-induced motion, in which case the above equation is simply iterated over. After this process is completed, then, for each element of the powder average, we bring these terms into the rotor frame, with angles $\Omega^{MF\rightarrow RF}=\left[\alpha^{MF\rightarrow RF},\beta^{MF\rightarrow RF},\gamma^{MF\rightarrow RF}\right]$.

$$A_{2,q}^{RF}=\sum_{q'=-2}^{2}D_{q',q}^{(2)}\left(\Omega^{MF\rightarrow RF}\right)A_{2,q'}^{MF} \qquad (11)$$

Finally, terms in the lab frame are given by time-dependent rotation of the rotor ($\omega_r t$) and rotation by the rotor angle ($\theta_r$).

$$A_{2,q}^{LF}=\sum_{q'=-2}^{2}e^{-iq'\omega_r t}d_{q',q}^{(2)}(\theta_r)A_{2,q'}^{RF} \qquad (12)$$

Note that if the spinning frequency is set to 0 Hz when initializing a simulation in SLEEPY, then the rotor angle is automatically set to zero, such that the final rotation has no effect and $A_{2,q}^{LF}=A_{2,q}^{RF}$. In SLEEPY, the powder average is always defined relative to the rotor, not the $B_0$ field, but by setting the rotor angle to zero, we align these two reference frames.

Then, we define $A_q^n=d_{n,q}^{(2)}(\theta_r)A_{2,n}^{RF}$, such that the total, lab-frame Hamiltonian for the corresponding interaction can be calculated according to

$$H(t)=\sum_{q=-2}^{2}(-1)^q\left[\sum_{n=-2}^{2}e^{-n\omega_r t}A_q^n\right]\hat{T}_{2,-q} \qquad (13)$$

where $\hat{T}_{2,-q}$ are components of the rank-2 spherical tensors. If the spin or spins involved in the corresponding interaction are both in the rotating frame, then terms for $q\neq0$ are discarded.

The powder average defines the set of angles, $\Omega^{MF-RF}$. SLEEPY may define the powder average via saved text files or via functions. Via text file, SLEEPY has the same powder averages available as in SIMPSON[1], including REPULSION[40], ZCW[41-43], and a Lebedev powder average for octahedral symmetry[44,45]. β-only powder averages are also

included, as well as a few single-angle averages ("alpha0beta0", "alpha0beta90", "alpha0beta45"). Note that these powder averages define only the α and β angles, and γ is then uniformly stepped between 0 and 2π. Via functions, SLEEPY includes a "grid" powder average, which creates a 3D grid over the $\alpha^{MF\rightarrow RF}$, $\beta^{MF\rightarrow RF}$, and $\gamma^{MF\rightarrow RF}$ angles with uniform spacing. SLEEPY also includes the Cheng, Suzukawa, and Wolfsberg powder average[43], referred to in SLEEPY as the "JCP59" powder average, which efficiently samples over all three angles simultaneously. Function-based powder averages may include arguments, for example, the grid powder average requires definition of the number of α, β, and γ angles, and the JCP59 powder average requires a quality factor, $q$ ("JCP59" is the default powder average, with $q=3$. If an integer is provided for the powder average, this will apply "JCP59" with $q$ set to that value). Note that powder averages include a "weight" factor that sums to 1, which is used to correctly average the different orientations together.

If a powder average does not define the γ angle, then γ will be uniformly stepped from 0 to 2π, with "n_gamma" angles, where "n_gamma" is provided to the ExpSys object, and is the same value as for number of steps taken to discretize the rotor period (in this case, the γ-COMPUTE algorithm[28] will be used with propagator caching, see "Caching and lazy execution"). Note that powder averaging over the γ-angle may be deactivated for these powder averages by setting "gamma_encoded = True" in ExpSys (although n_gamma steps will still be used throughout the rotor period). If a single-angle powder average is used, then "gamma_encoded" defaults to True (we assume the user only wanted a single angle), although the user may subsequently reset its value to False.

## Lab and rotating frames

By default, SLEEPY operates in the rotating frame, but for simulations where exchange should induce, e.g., longitudinal relaxation, terms typically dropped in the rotating frame approximation are important to induce relaxation, so we would rather work in the lab frame. SLEEPY can be set to operate in the rotating frame, lab frame, or rotating/lab frame can be specified spin-specifically. For the total Hamiltonian, $\hat{H}$, we obtain the rotating frame for all spins $i\in RF$ as

$$\hat{U}(t)=\prod_{i\in RF}e^{-i\omega_{0i}\hat{S}_{z,i}}$$

$$\widetilde{\hat{H}}=\overline{\hat{U}(t)\cdot\hat{H}\cdot\hat{U}^{\dagger}(t)}-\sum_{i\in RF}\omega_{0i}\hat{S}_{z,i} \qquad (14)$$

The term $\hat{U}(t)\cdot\hat{H}\cdot\hat{U}^{\dagger}(t)$ is time-dependent, where the line over it indicates an average over time, which then removes all time-dependent terms. Practically, removal of rotating terms is achieved

in SLEEPY by editing the spherical tensors, $\hat{T}_{2,q}$, according to whether an interaction is a spin-spin or spin-field interaction, whether it is a hetero- or homonuclear spin-spin interaction, and whether one or both spins are in the lab frame. The appropriate form of the rank-2 spherical tensors for each case is given in Table 1.

Note that applied radiofrequency fields in the rotating frame of the given spin are static, e.g., $H_{1i} = \omega_{1i}\hat{S}_{i,x}$, but if a spin is calculated in the lab-frame, then this term becomes time dependent, $H_{1i}(t) = 2\omega_{1i}\cos(\omega_{0i}t)\hat{S}_{i,x}$. This added time-dependence complicates application of radiofrequency fields considerably, because one would have to calculate minimally 2 steps per cycle of the Larmor frequency for every irradiated nucleus to correctly apply the field, amounting to ~10,000–100,000 steps per rotor cycle. SLEEPY supports continuous irradiation of one nucleus in the lab frame, which is achieved by taking two steps over one Larmor cycle, with a positive constant field applied in the first half of the cycle, and negative constant field applied in the second half ($\hat{H}_{1i}(t) = \pm\frac{\pi}{2}\omega_{1i}\hat{S}_{i,y}$, where $\frac{\pi}{2}$ provides the correct scaling to the main band, since undersampling of the cosine results in reduced field strength due to amplitude of higher frequency bands). This is used to create a propagator for one Larmor cycle, for one orientation of the rotor. This propagator is brought into its eigenbasis and propagated for a full step of the rotor cycle (i.e., for $\Delta t = 1/(n_{\gamma}\nu_r)$). The rotor position is then updated, and the process is repeated until a rotor period is completed. The number of steps per Larmor cycle can be increased, which could be necessary for homonuclear spin-systems to avoid higher frequency bands irradiating higher order transitions. In static mode, the same procedure is used, but we do not need to adjust for multiple rotor cycles.

## Eigenbasis propagation

In case a propagator will be repeatedly applied to a density matrix or raised to a power, it may be transformed into its eigenbasis for faster calculation. SLEEPY will automatically calculate the eigenbasis of a propagator (and store it) if that matrix is raised to a power, or if the propagator is used in the "rho.DetProp" function, which performs repeated detection and propagation with a propagator. For example, if $V$ diagonalizes $\hat{U}(t_0, t_0 + \Delta t)$, then signal at time $t_0 + n\Delta t$ becomes

$$\langle\hat{O}\rangle(t_0 + n\Delta t) = \hat{O}^{\dagger} \cdot \left(\hat{U}(t_0, t_0 + \Delta t)\right)^n \cdot \hat{\rho}(t_0)$$

$$D = V^{-1} \cdot \hat{U}(t_0, t_0 + \Delta t) \cdot V$$

$$\langle\hat{O}\rangle(t) = \left(\hat{O}^{\dagger} \cdot V\right) \cdot D^n \cdot \left(V^{-1} \cdot \hat{\rho}(t)\right) \tag{15}$$

The advantage here is that $D$ may be stored as only a vector, and so raising it to the $n^{\text{th}}$ power may be done far more quickly than raising the propagator matrix itself to the $n^{\text{th}}$ power.

Eigenbasis propagation is only valid if the given propagator is an integer number of rotor cycles (it is also valid if we are not spinning), and if a pulse sequence is applied, then we also require an integer number of sequence repetions. If a "sequence" is used for propagation in SLEEPY, instead of a propagator (sequences generate propagators in SLEEPY), one may also fit an integer number ($n$) of propagation steps within $m$ rotor periods. In this case, SLEEPY will construct a number of different propagators that are each an integer number of rotor cycles and an integer number of sequences, but starting at different time points within the $m$ rotor periods and $n$ sequences. Then, propagation in the eigenbasis (actually several eigenbases in this case) may still be applied, greatly reducing computational time. Note that the eigenbasis approach detailed here is closely related to the COMPUTE algorithm[29], although is carried out in Liouville space instead of Hilbert space.

SLEEPY performs some numerical cleanup in the eigenbasis. Some small numerical error can, in principle, produce eigenvalues marginally greater than one, but this would produce an infinitely growing system (manifesting, however, only on very long timescales). These eigenvalues are therefore reset to one. Second, if the system is thermalized, and the basis set is not reduced (see below for both procedures), then one eigenvalue must be exactly one, corresponding to the equilibrium of the system (thermal equilibrium, if no RF fields are applied). SLEEPY will search for the eigenvalue of $\hat{U}(t_0, t_0 + \Delta t)$ that is closest to one and set it to one, so that long simulations do not drift away from equilibrium due to numerical error. If a propagator is raised to infinity in SLEEPY ('U**np.inf' or 'U**"inf"', where 'np.inf' is infinity in the NumPy module and "inf" is a string) then all eigenvectors in $\hat{U}$ will be set to zero except the eigenvector corresponding to an eigenvalue of one. This is useful for finding the equilibrium of a density matrix resulting from a given propagator, where the equilibrium may not be thermal equilibrium in case radiofrequency irradiation is applied.

## Basis set reduction

SLEEPY can inspect the Liouville matrices, sequences, initial density matrix, and detection operators in order to reduce the size of the basis set required for propagation. The Liouvillian is first block-diagonalized, and then blocks that do not overlap with a non-zero element of both the initial density matrix and a detection operator are discarded. Discarding unused blocks results in significant speed up (depending on the size of the reduction possible).

Basis-set reduction is implemented by default when using the "rho.DetProp" function. If multiplying the density matrix by sequences or propagators, one has to first setup the reduced mode, using the "rho.ReducedSetup" function (SLEEPY otherwise has no way of knowing what propagators the user intends to use, and therefore does not know what elements of the basis are required). SLEEPY's basis set reduction is only applied where the reduction still results in an exact simulation, in contrast to Spinach's basis set reduction, which approximates the behavior of the simulation[30]. Note that SLEEPY uses the basis set of the Zeeman Hamiltonian (e.g., for a two-spin system: $\hat{S}_1^{\alpha}\hat{S}_2^{\alpha}$, $\hat{S}_1^{\alpha}\hat{S}_2^{-}$, $\hat{S}_1^{\alpha}\hat{S}_2^{+}$, $\hat{S}_1^{\alpha}\hat{S}_2^{\beta}$, $\hat{S}_1^{-}\hat{S}_2^{\alpha}$, $\hat{S}_1^{-}\hat{S}_2^{-}$, etc.). This allows basis set reductions for exact calculations to remove more matrix elements, than say a product operator basis ($\hat{S}_{1x}$, $\hat{S}_{1y}$, $\hat{S}_{1z}$, $\hat{S}_{1x}\hat{S}_{2x}$, etc.), but does not allow reduction according to spin-order[30], since terms such as $\hat{S}_1^{\alpha}\hat{S}_s^{\alpha}$ have a mixed spin-order ($\hat{S}_1^{\alpha}\hat{S}_2^{\alpha} = \frac{1}{4}\hat{E} + \frac{1}{2}\left(\hat{S}_{1z} + \hat{S}_{2z}\right) + \hat{S}_{1z}\hat{S}_{2z}$, where the first term has spin-order 0, the second term spin-order 1, and the third term spin-order 2).

## Exchange

SLEEPY implements exchange using two approaches. We start with the more general approach, in which we simply build $N$ separate spin systems with modulation of one or more interactions. Then, the individual $m \times m$ Liouville matrices for each spin-system are expanded into an $(N*m) \times (N*m)$ Liouville matrix, where each individual $m \times m$ Liouville matrix occupies a block on the diagonal of the larger matrix. An exchange matrix then connects all corresponding states of each spin-system in exchange. This is more easily understood if visualized, see Fig. 5.

This approach can be used with any number of different systems (always having the same spins, however), and with any valid (i.e., population conserving) exchange matrix. We may also use "Spin exchange". In this case, we only build a single spin-system, and allow spins within it to exchange. In this case, we need to add an exchange matrix that swaps the states of the spins in exchange. For example, if we have a two-spin system, then we would need to couple all states $\hat{I}_1^p\hat{I}_2^q$ to states $\hat{I}_1^q\hat{I}_2^p$. This is useful if, for example, we have a methyl group with protons exchanging between three sites or a phenylalanine ring

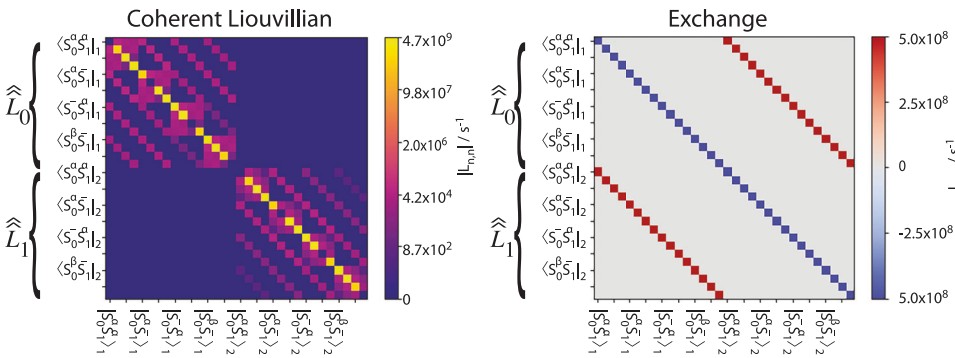

**Fig. 5 | Visualization of inclusion of exchange in SLEEPY.** To the left, we show how two Liouvillians, $\hat{L}_0$ and $\hat{L}_1$ have been inserted into a single matrix along its diagonal, and on the right we show a matrix that introduces exchange between corresponding states of the Liouvillian.

undergoing 180° flips. In both approaches to implementing exchange, we obtain a total Liouvillian that is the sum of the coherent and exchange contributions:

$$\hat{L} = \hat{L}_{\text{coh.}} + \hat{L}_{\text{ex.}} \tag{16}$$

## Tumbling models

Tumbling models can be useful for mimicking dynamics present in solution-state NMR due to tumbling in solution, using the "Setup-Tumbling" function in "Tools". We introduce tumbling by creating $N$ copies of the spin-system (ExpSys) where the anisotropic interactions will have new Euler angles appended for each copy, corresponding to reorientation in solution. The Euler angles are selected to mimic tumbling as described below, and the copies of Expsys are coupled together with an exchange matrix. Note that the "SetupTumbling" function assumes the user intends to simulate solution-state NMR, so will automatically set the powder average to "alpha0beta0" (one orientation along the $z$-axis), the rotor angle to 0, and the spinning frequency to zero. A flag, "solid", can be set to True to leave spinning and powder-averaging settings in place.

The user must specify which Euler angles to include for tumbling, and how many angles to use. β tumbling is always included. $\alpha$ is required if the initial set of interactions includes interactions with asymmetry, or that are not initially aligned along the $z$-axis. In solution NMR, $\gamma$ does not induce any shifts in spectral positions, and so plays no role in averaging away anisotropic interactions. It will induce relaxation processes in the lab frame, so for example, $T_1$ simulations should include $\gamma$ tumbling.

A quality factor ($q$) determines how many angles to average over. For β-only tumbling, $q$ will be multiplied by 10 to determine the number of β angles to be uniformly spaced between 0 and π. For two-angle tumbling, $q = 0$ will yield 3 sets of angles along the $x$, $y$, and $z$ axes (magic-angle hopping), $q = 1$ will yield tetrahedral hopping, and $q \geq 2$ will yield a REPULSION powder average[40], with increasing number of angles for increasing $q$ (Table 2). For three-angle tumbling, the "JCP59" powder is applied, where $q$ is passed as argument to the "JCP59" average. Weighting for the powder averages is also used in the construction of the exchange matrix in the next step.

For a set of Euler angles, we construct an exchange matrix to move between the angles. The three Euler angles are applied vectors starting from the $x$, $y$, and $z$ axes ($\vec{x}_j$, $\vec{y}_j$, $\vec{z}_j$). Then, for pairs of Euler angles, we calculate

$$D^2 = \arccos^2\left(\vec{x}_j \cdot \vec{x}_k\right) + \arccos^2\left(\vec{y}_j \cdot \vec{y}_k\right) + \arccos^2\left(\vec{z}_j \cdot \vec{z}_k\right) \tag{17}$$

which is the sum of squared angular distances between vectors. For each set of Euler angles, we then pick the nearest $n$ neighbors (Table 2), i.e., the $n$ smallest values of $D^2$. These are coupled in the exchange matrix, with $k_{j,k}^0 \propto 1/D^2$. A second term is added to $k_{j,k}$ to ensure correct equilibrium populations of the pairs of Euler angles by satisfying detailed balance. If $w_k$ and $w_j$ give the weighting of angles $k$ and $j$, we obtain

$$k_{j,k} = k_{j,k}^0 \left(1 + \frac{1 - w_k/w_j}{1 + w_k/w_j}\right)$$

$$k_{k,j} = k_{j,k}^0 \left(1 - \frac{1 - w_k/w_j}{1 + w_k/w_j}\right) \tag{18}$$

$k_{j,k}^0$ is defined as proportional to $1/D^2$. It is then scaled such that the inverse of the average rate constant resulting from this matrix matches a user-input correlation time.

$$\tau_c = \left(\sum_i A_i k_i\right)^{-1} \tag{19}$$

## Table 2 | Tumbling model details

| | β tumbling | 2-angle tumbling | 3-angle tumbling |
|---|---|---|---|
| Angles | Uniform spacing | Magic-angle hopping ($q = 0$) Tetrahedral hopping ($q = 1$)REPULSION | JCP59 |
| # of angles (vs. q) | — 1: 10 2: 20 3: 30 4: 40 5: 50 6: 60 | 0: 3 1: 4 2 :10 3: 20 4: 30 5: 66 6: 100 | — 1: x 2: 49 3: 99 4: 143 5: 199 6: 299 |
| # of neighbors (vs. q) | — 1: 2 2: 2 3: 2 4: 2 5: 2 6: 2 | 0: 2 1: 3 2 :5 3: 6 4: 6 5: 6 6: 6 | — 1: 15 2 :15 3: 15 4: 15 5: 15 6: 15 |

For each tumbling model (β only tumbling, 2-angle tumbling, and 3-angle tumbling), we first specify how a set of Euler angles is chosen, based on the powder average given in the "Angles" entry of the table. We also define a quality factor, $q$, which determines how many sets of Euler angles are used in the model. This is specified in the second entry of the table ("# of angles", where format is "q: Number of angles"). Finally, for each tumbling model, we define how many neighbors each set of Euler angles is coupled to. This is in the third entry of the table ("# of neighbors", where format is "q: Number of neighbors"). Note that higher $q$ values are supported than are listed in the table, but will lead to slow performance.

The rate constants $k_i$ are eigenvalues of the exchange matrix, excluding the zero eigenvalue (which must exist for a properly constructed exchange matrix). The $A_i$ are weightings for each rate constant, which sum to 1. They have been calculated assuming an interaction initially along the $z$-axis, according to the previously described procedure (SI of ref. [15]).

The "SetupTumbling" function will return a Liouvillian object. Note that the user may want to examine the tumbling model quality with the "Tools.L2A" function if multiple interactions are present. This will return $S^2$ for each interaction, the population, $p_{eq}$, of each angle, a list of correlation times, $\tau_c$, (the inverse of the exchange matrix eigenvalues), and a list of $A_i$ for each interaction. For each interaction, $S^2$ should approach zero if γ-tumbling is included. Correlation times corresponding to large values of $A_i$ should remain near the input $\tau_c$. Correlation times away from the input correlation time will add spectral density in the incorrect location. The extent of this distortion is proportional to the corresponding $A_i$.

With tumbling models, we encourage particular caution to ensure the user is including tumbling on the correct Euler angles, and to check for convergence of their simulations by increasing the quality factor, $q$, to ensure no significant changes. For 2-angle tumbling, $q = 0$ and $q = 1$ often return only qualitative results.

## Relaxation along the Cartesian coordinates

The first implementation of relaxation in SLEEPY is to define $T_1$ and $T_2$ relaxation of a single spin as occurring in the $z$-direction and in the $xy$-plane, respectively, via a one-spin random-field relaxation implementation[46]. This is applicable to a large number of simulations, and is computationally relatively simple. For $T_1$ relaxation, we calculate for spin $q$

$$\hat{\hat{L}}_q^+ = \hat{S}_q^+ \otimes \hat{1} - \hat{1} \otimes \left(\hat{S}_q^+\right)^T$$

$$\hat{\hat{L}}_q^- = \hat{S}_q^- \otimes \hat{1} - \hat{1} \otimes \left(\hat{S}_q^-\right)^T$$

$$\hat{\hat{L}}_q^{T_1} = --\frac{1}{4T_1}\left(\hat{\hat{L}}_q^+ \cdot \hat{\hat{L}}_q^- + \hat{\hat{L}}_q^- \cdot \hat{\hat{L}}_q^+\right) \tag{20}$$

$\hat{\hat{L}}_q^{T_1}$ is then the relaxation superoperator for $T_1$ on spin $q$. SLEEPY will remove diagonal terms that do not have corresponding off-diagonal terms from $\hat{\hat{L}}_q^{T_1}$. This takes away $T_2$ relaxation that necessarily comes with the $T_1$ relaxation. Doing this is, in principle, unphysical, but it allows the user to separate effects brought about specifically by $T_1$ relaxation in the absence of $T_2$ relaxation. SLEEPY produces a warning if the user does not add $T_2$ relaxation when using $T_1$ relaxation (although calculations will still run).

$T_2$ relaxation in the $xy$-plane is achieved by first calculating the $z$-superoperator and then taking its square, which will destroy coherences of spin $n$.

$$\hat{\hat{L}}_{q,z} = \hat{S}_{q,z} \otimes \hat{1} - \hat{1} \otimes \left(\hat{S}_{q,z}\right)^T$$

$$\hat{\hat{L}}_q^{T_2} = \frac{1}{T_2}\hat{\hat{L}}_{q,z} \cdot \hat{\hat{L}}_{q,z} \tag{21}$$

These Liouville matrices are added together, along with the coherent Liouvillian and exchange Liouvillian (if present).

$$\hat{\hat{L}}_{\text{relax}} = \sum_q \hat{\hat{L}}_q^{T_1} + \hat{\hat{L}}_q^{T_2}$$

$$\hat{\hat{L}} = \hat{\hat{L}}_{\text{coh.}} + \hat{\hat{L}}_{\text{ex.}} + \hat{\hat{L}}_{\text{relax}} \tag{22}$$

## Relaxation in the eigenbasis

Relaxation along the cartesian coordinates is sufficient for most calculations, because the strong external magnetic field brings the eigenbasis of the Hamiltonian into the direction of the field. However, it can be that the eigenbasis is tilted away from the $z$-axis, for example for an electron with significant $g$-anisotropy, or for spins that are strongly mixed, as can be the case for a homonuclear spin system or an electron-nuclear system with strong hyperfine coupling. In this case, if we apply $T_2$ relaxation to a tilted spin as described in the previous section, it will mix into the $T_1$ relaxation, therefore accelerating $T_1$ significantly, which may not be consistent with the behavior desired by the user. It may also cause thermalization to fail. Therefore, SLEEPY includes the option to implement $T_1$ and $T_2$ relaxation in the eigenbasis of the Hamiltonian. In this case, $T_1$ relaxation is defined as an exchange between eigenstates of the Hamiltonian, and $T_2$ only destroys coherences between the eigenstates.

Supposing we want to add $T_1$ relaxation to spin $n$, we start with $\hat{\hat{L}}_n^{T_1}$ as in Eq. (20), and bring this into the eigenbasis of the Hamiltonian. To achieve this transformation, we first diagonalize the Hamiltonian:

$$\hat{D} = \hat{V}^\dagger \cdot \hat{H} \cdot \hat{V} \tag{23}$$

$\hat{V}^\dagger$ indicates the complex conjugate of $\hat{V}$, which is equal to its inverse. If $\hat{\hat{L}}_{\text{coh.}}$ is the coherent Liouvillian resulting from Eq. (2), then it can be diagonalized using the right and left superoperators of $\hat{V}$ and $\hat{V}^{-1}$.

$$\hat{\hat{V}}_L = \left(\hat{1} \otimes \hat{V}\right) \cdot \left(\hat{V}^\dagger \otimes \hat{1}\right)$$

$$\hat{\hat{V}}_L^{-1} = \left(\hat{V}^\dagger \otimes \hat{1}\right) \cdot \left(\hat{1} \otimes \hat{V}\right)$$

$$\hat{\hat{D}}_L = \hat{\hat{V}}_L^{-1} \cdot \hat{\hat{L}}_{\text{coh.}} \hat{\hat{V}} \tag{24}$$

We apply this transformation to $\hat{\hat{L}}_n^{T_1}$, and select the largest $m$ off-diagonal elements, where $m$ is the expected number of $T_1$ transitions. This finds the transitions closest to the $T_1$ transitions that would occur in a non-tilted/non-mixed system. These are all set to $1/(2{*}T_1)$ multiplied by the phase of the corresponding element of $\hat{\hat{L}}_n^{T_1}$ in the diagonal frame, and the diagonal elements are set to $-1/(2{*}T_1)$. Note this approach is not recommended for spins with $S > 1$ where multiple transition rates are expected to have different values for the same nucleus. The resulting matrix is then transformed back into the original frame.

$T_2$ relaxation is added in the eigenbasis by calculating

$$\hat{\hat{V}}_L^{-1} \cdot \left(\hat{\hat{L}}_{n,x}^2 + \hat{\hat{L}}_{n,y}^2 + \hat{\hat{L}}_{n,z}^2\right) \cdot \hat{\hat{V}}_L \tag{25}$$

Only diagonal elements are kept from this matrix, and diagonal elements occurring in the $T_1$ relaxation are also discarded. The resulting diagonal matrix is then transformed back into the original frame to obtain the $T_2$ relaxation matrix for the $n$th spin.

Note that it may not always make sense to apply $T_2$ in the eigenbasis for two spins with mixed eigenstates due to a coupling. The reason is as follows: the stochastic field acting on one spin should

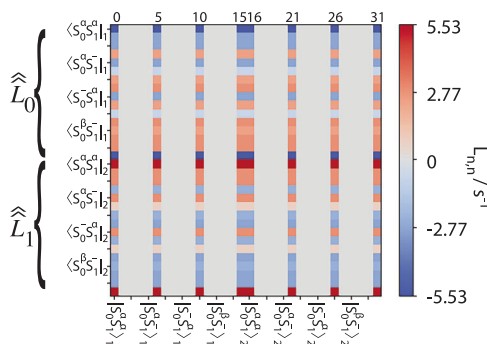

**Fig. 6 | Visualization of the dynamic thermalization matrix.** Here, a 2-spin-1/2 system undergoing 2-site exchange yields a 32 × 32 Liouvillian. The columns are each a copy of the product $\hat{\hat{L}}.\rho_{eq}$ (the matrix product of the full Liouvillian and the equilibrium density matrix). Columns 0, 5, 10, and 15 correspond to the first identity matrix (i.e., a 4 × 4 identity matrix extended into a single column), and this is repeated for the second system in exchange, so columns 16, 21, 26, and 31 correspond to the second identity matrix. Note that $\hat{\hat{L}}.\rho_{eq}$ has been scaled up (and the identity in $\hat{\rho}_{eq}$ scaled down) to yield better numerical stability.

influence all coherences between eigenstates that include that spin, although to different degrees. Applying $T_2$ only to the coherences corresponding most strongly to the given spin will allow other coherences to survive indefinitely, although ultimately the same stochastic field should have an influence on all coherences involving that spin. This can be addressed in some cases by simply applying $T_2$ along the $xy$-plane rather than using a tilted/mixed frame. In other cases, where relaxation in the eigenbasis is required, one can just apply $T_2$ relaxation to both mixed spins.

### Lindblad thermalization
SLEEPY uses Lindblad thermalization to bring a spin system back towards its thermal equilibrium for most simulations, except where we want to thermalize a system where relaxation is brought about by exchange in the system (next section). Bengs and Levitt[27] require that if our non-thermalized relaxation matrix is given by $\hat{\hat{L}}_{relax}$ (referred to as the adiabatic contribution), then the thermalized relaxation matrix is given by

$$\hat{\hat{L}}_{relax}^{\theta} = \hat{\hat{L}}_{relax}^{\theta,ad} + \hat{\hat{L}}_{relax}^{\theta,na}$$

$$\hat{\hat{L}}_{relax}^{\theta,ad} = \hat{\hat{L}}_{relax} \tag{26}$$

Superscripts "ad" and "na" refer to adiabatic and non-adiabatic contributions to the relaxation. The adiabatic contributions yield a symmetric matrix, and the non-adiabatic contributions yield an anti-symmetric matrix. The latter equation above indicates that the symmetric part of the relaxation matrix should not be modified when thermalizing the system. For off-diagonal matrix elements of $\hat{\hat{L}}_{relax}^{\theta}$ between states $r$ and $s$, it is required that the forward and reverse rate constants have the following relationship:

$$W_{r \to s}^{\theta} = W_{s \to r}^{\theta} \exp(-\beta_{\theta} \omega_{sr})$$

$$\beta_{\theta} = \frac{\hbar}{k_B T} \tag{27}$$

Given $W_{r \to s}^{\theta,ad} = W_{s \to r}^{\theta,ad}$, we may solve for $W_{r \to s}^{\theta,na} = -W_{s \to r}^{\theta,na}$

$$W_{r \to s}^{\theta,ad} + W_{r \to s}^{\theta,na} = (W_{r \to s}^{\theta,ad} + W_{r \to s}^{\theta,na}) \exp(-\beta_{\theta} \omega_{sr})$$

$$W_{r \to s}^{\theta,na} = W_{r \to s}^{\theta,ad} \frac{1 - \exp(-\beta_{\theta} \omega_{sr})}{1 + \exp(-\beta_{\theta} \omega_{sr})} \tag{28}$$

This gives the off-diagonal elements. The diagonal elements of the non-adiabatic matrix must conserve populations in the exchange matrix, so that $W_{r \to r}^{\theta,na} = -W_{r \to s}^{\theta,na}$ and $W_{s \to s}^{\theta,na} = -W_{s \to r}^{\theta,na}$.

We still must define $\omega_{sr}$, which is the difference $\omega_s - \omega_r$. $s$ and $r$ index diagonal elements of the Liouvillian, which correspond to the matrix elements of the Hamiltonian. If $s$ corresponds to element $(p,q)$ of the Hamiltonian, then $\omega_s = \left(\omega_{p,p} + \omega_{q,q}\right)/2$, that is, it is the mean of the corresponding diagonal elements of the Hamiltonian. This is technically only fully correct if we have diagonalized the Hamiltonian, because otherwise the off-diagonal elements would modify the energy levels. If we use relaxation in the eigenbasis, as described above, then energy levels are indeed extracted from the diagonal elements of the diagonalized Hamiltonian. If relaxation is not implemented in the eigenbasis of the Hamiltonian, then we use the diagonal elements of the non-diagonalized Hamiltonian as good approximation. Note that if relaxation in the eigenbasis is combined with Lindblad thermalization and spinning, then the thermal equilibrium is effectively changed at every step in the rotor period and also varies among crystallites in the powder average. For most simulations, this may not make a significant difference but does have an impact on experiments such as the PCS.

Note that in the tilted or mixed frame, $W_{r \to s}^{\theta,ad}$ may be complex. Then, for the off-diagonal elements, $W_{r \to s}^{\theta,na}$, phase information should be retained, but for diagonal elements, $W_{r \to r}^{\theta,na}$, the absolute value should be used (while retaining the sign arising from $1 - \exp\left(-\beta_{\theta} \omega_{sr}\right)$).

### Dynamic thermalization
Longitudinal relaxation in SLEEPY may come from user-defined settings of $T_1$, which are then thermalized via Lindblad thermalization. However, exchange processes also may induce $T_1$, where it is not straightforward to correctly thermalize the system via Lindblad thermalization, since we do not have the relaxation rate constants directly. For this case, SLEEPY has the "Dynamic thermalization" method (`L.add_relax("DynamicThermal")`). To understand this approach, we start with the equation of motion for the density matrix *without* any radio-frequency irradiation or thermalization.

$$\frac{d}{dt}\hat{\rho}(t) = \hat{\hat{L}} \cdot \hat{\rho}(t) \tag{29}$$

If we insert $\hat{\rho}(t) = \hat{\mathbf{0}}$ into this equation, we necessarily get $\frac{d}{dt}\hat{\rho}(t) = 0$, but we may modify the equation as follows:

$$\frac{d}{dt}\hat{\rho}(t) = \hat{\hat{L}} \cdot \hat{\rho}(t) - \hat{\hat{L}} \cdot \hat{\rho}_{eq}$$

$$\hat{\rho}_{eq} \propto \exp\left(-\frac{\hat{H}}{k_B T}\right), \sum_i \hat{\rho}_{eq}^i = 1 \tag{30}$$

In this case, if $\hat{\rho}(t) = \hat{\rho}_{eq}$, then $\frac{d}{dt}\hat{\rho}(t) = 0$, yielding a system that approaches thermal equilibrium[26]. Note that this does not yield correct thermalization of coherences[27], and so will not correctly reproduce effects like the PCS.

While Eq. (30) yields the desired relaxation towards $\hat{\rho}_{eq}$, it is not easily implemented as such in SLEEPY, since Eq. (3) is no longer the solution to this equation. We need to insert the $\hat{\hat{L}} \cdot \hat{\rho}_{eq}$ correction term

**Table 3 | SLEEPY Benchmarks**

| Benchmark | Time/s | Size (before/after reduction) | #Orientations | Caching[a] (calculated/ recycled) | Memory usage (approx./GB) | CPU usage (approx.) |
|---|---|---|---|---|---|---|
| $R_{1\rho}$ (series) | 0.24 | 32/16 | 49 | 1470/0 | <1 | _[b] |
| $R_{1\rho}$ (parallel) | 0.27 | 32/16 | 49 | 1265/0 | <1 | _[b] |
| REDOR (series) | 18 | 48/48 | 600 | 13800/45000 | <1 | 500% |
| REDOR (parallel) | 26 | 48/48 | 600 | 13791/43582 | <1 | 1000% |
| PCS with MAS (series) | 18 | 16/16 | 299 | 8970/0 | <1 | 100% |
| PCS with MAS (cached) | 2.0 | 16/16 | 299 | 0/8970 | <1 | 125% |
| Water hop (4 spin, series) | 3.9 | 256/80 | 49 | 4900/0 | <1 | 300% |
| Water hop (4 spin, parallel) | 4.1 | 256/80 | 49 | 4900/0 | <1 | 1000% |
| Water hop (4 spin, parallel cached) | 1.1 | 256/80 | 49 | 0/4900 | <1 | _[b] |
| Water hop (5 spin, series) | 65 | 1024/280 | 49 | 4900/0 | ~9/~1.5[d] | 600% |
| Water hop (5 spin, parallel) | 60 | 1024/280 | 49 | 4900/0 | ~12/~5[d] | 1150% |
| Water hop (5 spin, parallel cached) | 21 | 1024/280 | 49 | 0/4900 | ~12 | 1150% |
| Water hop (6 spin, series) | 2316[c] | 4096/1008 | 49 | 4900/0 | ~20[e] | 600% |
| Water hop (6 spin, parallel) | 1863[c] | 4096/1008 | 49 | 4900/0 | ~20[e] | 1150% |

Each row of the table provides benchmark results for the given simulation (simulation code provided in "Supplementary Software 1.py"). Time is the wall time (Mac Mini 3.2 GHz 6-Core Intel Core i7, 32 GB RAM), size is the size of one dimension of the Liouville matrix where the total size and the size after matrix reduction is given, # of orientations is the number of angles used in the powder average, Caching is the number of rotor-step propagators calculated in total vs. the number that could be recycled, Memory usage is an estimate of the memory based on the "Activity Monitor", and CPU usage is an estimate also based on the "Activity Monitor". Most simulations have been run in both series and parallel, specified in the leftmost entry of the table. Pseudocontact shift (PCS) and water-hop simulations were run twice, where the second simulation could use cached propagators from the first simulation, to demonstrate speedup from the cache.
[a]For parallel simulations, the cache count is stored in shared memory and can miss counts due to simultaneous editing by two processes; therefore counts are potentially lower for the parallel simulations.
[b]Simulations terminate before CPU/memory usage can be assessed.
[c]Includes 109 s time for setup of the exchange (68 s) and spin-diffusion matrices (41 s).
[d]Memory usage is assessed with and without caching.
[e]Propagator caching was deactivated for these calculations to reduce memory usage. We do not expect propagator caching to be beneficial in these calculations.

(a 1D vector) into the Liouvillian itself. This may be done by noting that $\hat{\rho}_{eq}$ originates from flattening a square matrix into a single vector. The identity matrix of the same shape is then included in $\hat{\rho}_{eq}$, and is unmodified under evolution with the Liouvillian. We may couple the correction vector to this identity in the density matrix by inserting $\hat{\hat{L}} \cdot \hat{\rho}_{eq}$ along each column corresponding to the identity. For example, for a $4 \times 4$ Hamiltonian, $\hat{\hat{L}} \cdot \hat{\rho}_{eq}$ should be inserted in the 0th, 5th, 10th, and 15th columns of the Liouvillian, which line-up with the identity found in the density matrix. This is illustrated below in Fig. 6.

This approach has the drawback that the lab frame Zeeman Hamiltonian and fast exchange processes yield very large terms in the Liouvillian, whereas the Dynamic Thermalization matrix yields very small terms. This can increase the chances of numerical error causing thermal equilibrium to not be properly achieved. We can partially compensate this by scaling up the terms in $\hat{\hat{L}} \cdot \hat{\rho}_{eq}$ and scaling down the identity in $\hat{\rho}(t)$ by the same factor to yield improved numerical stability, but cases still exist where thermalization fails (especially for larger systems with fast motion, $\tau_c < 10^{-10}$ s, see tutorial: Solution NMR/$T_1$ limits). Generally, the user should check that nuclei achieve the appropriate thermal equilibrium values (compare $z$-magnetization after a long evolution to stored equilibrium values in "ex.Peq").

## Caching and lazy execution

SLEEPY uses caching and lazy execution to speed up computation. When the Liouvillian for a particular element of the powder average is accessed, then the corresponding $\hat{L}_n$ are calculated and cached until that element is no longer accessible in memory. Propagators for a given step in the rotor period are cached in the full Liouvillian cache ("L._PropCache"). These will be recycled if multiple propagators are created from the Liouvillian, but are also used to reduce calculations over the powder average. Specifically, if we discretize a rotor period into $M$ steps with length $\Delta t = 1/(M\nu_r)$ and change the $\gamma$-angle for each

step using $\Delta\gamma = 2\pi/M$, we find that some propagators may be recycled[47], if the following condition is met:

$$\hat{\hat{U}}^{\alpha, \beta, \gamma}(m\Delta t, (m+1)\Delta t) = \hat{\hat{U}}^{\alpha, \beta, \gamma'}(m'\Delta t, (m'+1)\Delta t)$$

if

$$(\gamma - \gamma')/\Delta\gamma \bmod M = (m' - m) \bmod M \qquad (31)$$

Note that this is only valid if the applied radiofrequency field is the same at both times. Then, SLEEPY will keep multiple field-dependent caches for this purpose (up to 10 field settings by default).

When a propagator object is created (the above discussion references propagator matrices calculated along the way to generating a propagator object), it is not actually calculated until needed (lazy execution). Calculation occurs when it is either multiplied by another propagator or density matrix, or raised to a power. Once calculated, the result is stored and used for any further calculations with the propagator. If propagation in the eigenbasis is used, then the eigenvalues and eigenvectors are also cached.

Finally, when using orientation-specific relaxation (L.add_relax(….,OS=True)), relaxation matrices are cached when they are created. Note that the sl.Defaults['cache'] can be used to deactivate some caching, but is only applied to propagators for a given step in the rotor period and for relaxation matrix caching. Caching of the $\hat{L}_n$ and of propagator objects and their eigenvalues is always active.

## Benchmarks

We provide performance times as well as other information for a number of simulations. Benchmarks have been performed on a Mac Mini 3.2 GHz 6-Core Intel Core i7 with 32 GB of memory, running Sequoia 15.4.1, with Python version is 3.11.13, NumPy 1.24.3[48], SciPy

1.10.1[49], and Multiprocess 0.70.15. NumPy and SciPy are configured to use Intel's MKL BLAS and LAPACK linear algebra libraries. Note that the choice of linear algebra libraries has a significant impact on performance. Table 3 summarizes the results, including total computational time, Liouville matrix sizes before and after matrix reduction (the reduced size usually being more relevant for computational time), number of powder orientations, caching usage (number of calculated propagators for rotor cycle steps/number of recycled propagators), and approximate peak memory and CPU usage. Memory usage and CPU usage are estimated only approximately using the Activity Monitor utility. They correspond to the approximate peak usage during each simulation. Benchmark code is included as "Supplementary Software 1.py" as a Supplementary Software file, and is also found at http://sleepy-nmr.org/html/Chapter7/Ch7_SleepyBenchmark.html.

When considering the SLEEPY benchmarks, a few important considerations should be made. First, parallelization of the powder average is not always beneficial. NumPy and SciPy already use parallel processing in their libraries, and so the gain in speed from parallelization is already reduced, and one further loses some efficiency due to overhead requirements of the parallel processes (activate with `sl.Defaults['parallel']=True`). Second, we see that some simulations are able to use a significant fraction of cached propagators (e.g., REDOR), whereas others cannot ($R_{1\rho}$). This depends also if multiple propagators are calculated and on the powder average used ("JCP59" only uses caching when multiple propagators are calculated). Then, for large simulations (e.g., water hop with 5/6 spins), it may be beneficial to deactivate caching (`sl.Defaults['cache']=False`).

Water hop simulations (adapted from Vinod-Kumar et al.[50]) can be used to assess scaling as a function of the number of spins. Between the 5- and 6-spin simulations, we see computational time increase by a factor of $2316\,\text{s}/65\,\text{s}=35.6$. Strassen's algorithm predicts a scaling exponent of 2.81 for matrix multiplication, yielding $(1008/280)^{2.81}=36.6$. Increase in computational time of smaller spin systems tends to be less due to other calculations contributing a greater portion to the overall computational time. At present, SLEEPY is limited to about 6 spin-1/2 nuclei.

## Data availability
The simulated data generated in this study have been provided as a Source Data file provided with this article in the file SourceData.zip. Source data are provided with this paper.

## Code availability
The SLEEPY source code is available under the GNU General Public License version 3. It has been permanently archived with Zenodo (DOI: 10.5281/zenodo.14886944)[51]. It is also available via GitHub, at https://github.com/alsinmr/SLEEPY and at PyPi at https://pypi.org/project/sleepy-nmr/.

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

## Acknowledgements

This project was funded through the Deutsche Forschungsgemeinschaft (DFG) grant 450148812 (A.A.S.), the European Social Funds (ESF) and the Free State of Saxony Junior Research Group UniDyn Project No. SAB 100382164 (A.A.S., K.Z.). We would also like to extend special thanks to Matthias Ernst for many helpful discussions on approaches to simulating relaxation and dynamics.

## Author contributions
A.A.S. has designed the SLEEPY code with support from K.Z. A.A.S. has written the SLEEPY tutorial and K.Z. has tested the code and tutorial. A.A.S. and K.Z. have written the manuscript.

## Funding

## Competing interests
The authors declare no competing interests.
