## [Transparent Peer Review file · Nature Communications]

SLEEPY: A comprehensive Python module for simulating relaxation and dynamics in Nuclear Magnetic Resonance

Corresponding Author: Dr Albert Smith

Version 0:

Reviewer comments:

Reviewer #1

(Remarks to the Author)

The paper presents a new software package for general purpose simulations of dynamic processes in NMR. As argued by the authors, the main strength of NMR spectroscopy is its ability to characterize motion, molecular flexibility, complexation, and thus to reveal (bio)molecular function. Easy and simple access to numerical simulations is indispensable in data evaluation and method developments. The presented SLEEPY package can influence the field in similar way that SIMPSON did for solid-state NMR, becoming standard simulation software used in almost every lab. There are big opportunities but also great responsibility that simulations are right. The main challenge I see, is in educating users. SLEEPY provides simple environment for quick setup but it hides technical details from the user. There are many difficult decisions to be made (presumably by the user, or the software itself?) to obtain correct and meaningful results. Who will decide that the approximations are correct, time-steps are small enough, powder averages sufficient, relaxation models reflect the reality, etc? At the end, it is the user's responsibility. But would the user think about it when he/she is not directly exposed to such decisions?

The paper certainly deserves publication, but the warning expressed above should be clearly stated in the software presentations. The authors should also consider my other comments below.

In introduction, SLEEPY should be compared to other existing software. We learn about SIMPSON but there are other general purpose NMR simulators, namely SPINACH that offers the very same capability of simulations in Liouville space, including relaxation as well as dynamics. Note also SPINEVOLUTION and the recent python package MRSIMULATOR.

I have not found any information about computational demands, no assessment how quick the simulations are (wall time of calculations), what computers are best to use (laptops, grids, supercomputers?), and what are the limits regarding system sizes (number of spins, number of exchange sites,...) , parallelization, ...

Numerical methods are described at the end of the paper. It is OK, but some general information would be useful already when presenting example simulations, and what are the approximations made at those calculations.

Lab frame and rotating frame – these terms should be explained, what lab frame? Does it address the spin space (i.e. spin operator interactions in full, prior the secular approximation) or the 3D space of anisotropic interactions?

Example in Figure 4 – “tumbling model with 10 sets of Euler angles”. The highlighted code does not indicate anything about 10 sets. This is just an example of how important approximations are hidden from the user. Are 10 sets good enough to mimic isotropic reorientation of a molecule? How are those 10 orientations selected? How can the user influence this?

There is no discussion of powder averaging. On page 12, n_{γ} is used but not described how to set it. In my opinion, insufficient references are used, for example the COMPUTE algorithm implemented here was described by Eden and Levitt before, γ -COMPUTE was introduced by Nielsen et al. ...

For calculations, all operators should be expressed in some basis. What is it for SLEEPY? Is it the eigenbasis of the Zeeman Hamiltonian or a basis of product operators? It becomes relevant when describing exchange operators L_{exc} as well as frequencies ω_{SR}

Relaxation models are extremely difficult to set properly. When to use relaxation along cartesian coordinates and when in the eigenbasis? What type of thermalization to use? What is ρ_{eq} introduced in Eq.21? How to get it correctly? Is the approximation that all coherences decay with the same time constant always valid? And how to recognize numerical errors arising from the "dynamic thermalization" noted on page 20? As I said in the beginning, many difficult decisions are hidden...

Minor discrepancies in Figure 1, (A) – $n=4000$ in the code but 6000 steps in the comment, (B) current time 88888,89 microsecond but 100 ms in the comment.

Figure 2 – seqx seqmx in the legend, cwx and cwmx in the figures

(Remarks on code availability)

Reviewer #2

(Remarks to the Author)

The SLEEPY program to simulate complex NMR pulse sequences, including relaxation and chemical exchange, will be a very valuable resource for NMR spectroscopists as they develop new and improved experiments. Thus I believe that this program will be of use and should be published. My concern is where. Other, also powerful programs are published in JMR and I think that this is the best place for this work as well.

(Remarks on code availability)

Reviewer #3

(Remarks to the Author)

The authors propose a new python based software for simulating NMR relaxation and dynamics. The paper is well written but it is a specialized paper in NMR. It should be submitted to "Journal of Magnetic Resonance". It is not most suitable for nature communications

(Remarks on code availability)

Version 1:

Reviewer comments:

Reviewer #1

(Remarks to the Author)

The revised version of the manuscript addresses all my previous comments sufficiently. The work can be considered for publishing.

(Remarks on code availability)

Reviewer Response

Thanks to all reviewers for the kind remarks and helpful comments.

Please see our responses below. We've responded to reviewer 2 and 3 first, as one response, since the comments are very similar.

Note that lines/pages refer to the marked version of the paper, not the clean version.

Reviewer #2 (Remarks to the Author):

The SLEEPY program to simulate complex NMR pulse sequences, including relaxation and chemical exchange, will be a very valuable resource for NMR spectroscopists as they develop new and improved experiments. Thus I believe that this program will be of use and should be published. My concern is where. Other, also powerful programs are published in JMR and I think that this is the best place for this work as well.

Reviewer #3 (Remarks to the Author):

The authors propose a new python based software for simulating NMR relaxation and dynamics. The paper is well written but it is a specialized paper in NMR. It should be submitted to "Journal of Magnetic Resonance". It is not most suitable for nature communications

Thanks for the positive reception of our work. The question of where to publish this work is certainly relevant, and something we considered. It's of course very important to us that this work is shared within the magnetic resonance community, although I do think that we will reach that community by publishing in Nature Communications.

The other side of this question is whether it is interesting outside of the core NMR community. We have been mainly focused on dynamics for a number of years now, and while NMR is remarkably powerful for characterizing dynamics, especially for timescale-specific analysis, we find it is often very difficult to give a clear interpretation of experimental results without a second method. Our group typically uses molecular dynamics simulation (MD) as the supporting method, or sometimes also as the main method with NMR providing support. In either case, it means that many of these studies become multi-disciplinary. Then, when, for example, an MD expert tries to understand what role exactly T_1 , CEST, REDOR, exchange, etc. plays in a study, they can potentially use SLEEPY to experiment with critical parameters and understand the experimental response. We don't expect the non-NMR users to build their simulations from scratch, but this is where online sharing and a tutorial has the potential to play a major role, because one then has the opportunity to experiment with a few parameters for a pre-built simulation. I am cautiously optimistic that researchers make shareable code with SLEEPY (or whatever other NMR-based software that is developed) that they can include links to in their publications. I think with existing programs (SIMPSON, Spinach), within the NMR community, one may try to reproduce published simulations, since it is not uncommon to already have these programs already installed. However, we think that outside this community, it is less likely that simulations are investigated. But with SLEEPY, potentially researchers outside the NMR field can also try simulations since they can be already run online without any special setup. So, in this way we hope to target a broader audience by publishing in Nature Communications.

Reviewer #1 (Remarks to the Author):

The paper presents a new software package for general purpose simulations of dynamic processes in NMR. As argued by the authors, the main strength of NMR spectroscopy is its ability to characterize motion, molecular flexibility, complexation, and thus to reveal (bio)molecular function. Easy and simple access to numerical simulations is indispensable in data evaluation and method developments. The presented SLEEPY package can influence the field in similar way that SIMPSON did for solid-state NMR, becoming standard simulation software used in almost every lab. There are big opportunities but also great responsibility that simulations are right. The main challenge I see, is in educating users. SLEEPY provides simple environment for quick setup but it hides technical details from the user. There are many difficult decisions to be made (presumably by the user, or the software itself?) to obtain correct and meaningful results. Who will decide that the approximations are correct, time-steps are small enough, powder averages sufficient, relaxation models reflect the reality, etc? At the end, it is the user's responsibility. But would the user think about it when he/she is not directly exposed to such decisions?

The paper certainly deserves publication, but the warning expressed above should be clearly stated in the software presentations. The authors should also consider my other comments below.

Thanks for this important point. Our goal was to simultaneously get users started quickly and reduce the amount of code, but also to allow access to technical details. For example, we give access to all the matrices (Hamiltonians, Liouvillians, Propagators), including built-in plotting functions but also the raw numerical matrices, broken into their components (rotating components, relaxation matrices, exchange matrices). The latter point is perhaps not emphasized strongly enough.

Page 11, line 245-247: We point out that various settings can be plotted, and matrices are user-accessible.

However, in SLEEPY almost everything has a default value that the user does not need to touch for some simulations, but may need to adjust for others. Some things are sensible: the powder average will automatically be removed if no anisotropic interactions are included or a tumbling model is applied, and the rotor angle goes to 0 if no spinning is included to get better powder average performance. But, if a powder average is required, then the default powder average performs well for experiments such as $R_{1\rho}$, T_1 , etc., but poorly for REDOR, for example. Since the user may not see where the powder average is set, they may not adjust it. Other considerations are when to use lab frame calculations, when to use tilted-frame relaxation, etc.

Page 11-12, line 243-279: We have added a section to discuss "Complexities of Simulation" which points out the dangers of having default values and that the users need to check their settings. As examples, we discuss powder averaging and choice of relaxation model.

In introduction, SLEEPY should be compared to other existing software. We learn about SIMPSON but there are other general purpose NMR simulators, namely SPINACH that offers the very same capability of simulations in Liouville space, including relaxation as well as dynamics. Note also SPINEVOLUTION and the recent python package MRSIMULATOR.

Page 2, Line 28-29: We have added references to SpinEvolution and SPINACH.

MRSimulator seems to me to be more special-purpose software (it is nicely done, but cannot handle arbitrary pulse sequences). I think if we cite every piece of Python software, we would have far too many. However, it does check all the boxes for sharing-features by being implemented in Python, so we've added a reference to it in that context.

Page 12, line 288-290: We have added a citation to MR Simulator

I have not found any information about computational demands, no assessment how quick the simulations are (wall time of calculations), what computers are best to use (laptops, grids, supercomputers?), and what are the limits regarding system sizes (number of spins, number of exchange sites,...) , parallelization, ...

We've added a set of benchmarks to the Methods, along with a discussion.

Page 30-31: We've added a section "Benchmarks" that looks at a number of simulations, their wall time, and discusses in detail the factors affecting computational efficiency.

Page 3, line 119-121: Reference to the benchmarks in the main text.

Numerical methods are described at the end of the paper. It is OK, but some general information would be useful already when presenting example simulations, and what are the approximations made at those calculations.

Thanks for the suggestion. We've added a section "Simulation basics" to the main text to introduce the basic functionality of SLEEPY and how it works, without getting into too much detail (which is still provided, now in the SI).

Note that SLEEPY is not using approximations of the Liouvillian, except for the optional rotating-frame approximation. Basis set reduction is only applied where states of the system are completely decoupled, and so is not an approximation beyond the initial rotating-frame approximation (in the lab frame, the systems don't typically reduce). Added relaxation is of course only a model; this is also discussed.

Page 3-5, line 69-121: A discussion of the basics of simulation in SLEEPY is now included.

Page 5, line 115-117: A brief comment on the basis set reduction is added to clarify that this is not an approximation

Page 11, 256-275: Discussion of strengths and limitations of the relaxation models is now discussed.

Lab frame and rotating frame – these terms should be explained, what lab frame? Does it address the spin space (i.e. spin operator interactions in full, prior the secular approximation) or the 3D space of anisotropic interactions?

The rotating frame/lab frame addresses the spin space, i.e. the inclusion or not of the full set of spin operators (in SLEEPY, implemented as spherical tensors). I'm not quite sure what else this would refer to, except that we do refer to the lab frame and *rotor* frame for the definition of anisotropic interactions (admittedly similar terminology). In any case, to clarify, in the main text, we add reference to the secular and pseudo-secular approximations, and also define the lab, rotating, and mixed frames carefully in the SI.

Page 4, line 98-104: Discusses lab, rotating, and mixed frames, and their relevance for dynamics simulations.

Page 17-18, lines 402-431: Mathematically defines the rotating frame Hamiltonian, and tabulates the spherical tensors for a spin interaction with both spins in the lab frame, both in the rotating frame (including hetero- vs. homonuclear), and in mixed frames.

Example in Figure 4 – "tumbling model with 10 sets of Euler angles". The highlighted code does not indicate anything about 10 sets. This is just an example of how important approximations are hidden from

the user. Are 10 sets good enough to mimic isotropic reorientation of a molecule? How are those 10 orientations selected? How can the user influence this?

My apologies for the detour that is about to follow: the reviewer's question is eventually answered below:

Revisiting the tumbling model has been fairly productive, and we've expanded/corrected our code to account for some issues and also to make tumbling more efficient in some cases.

So, first, for a tumbling model, one has the potential to tumble on all three Euler angles. At the time of submission, we were always fixed to tumbling over one or two angles. This is often sufficient, but now we have the ability to turn on and off tumbling on the alpha angle and on the gamma angle. For example, if we only have symmetric tensors that are aligned, then alpha tumbling has no effect, and so it's faster to omit it. Second, if we're interested in just the averaging of interactions (e.g. solution-state PCS), then the gamma angle has no effect (gamma motion does lead to relaxation if it's on the right timescale). However, we could need both in case we have asymmetric tensors, or unaligned tensors, and we're interested in looking at longitudinal relaxation. So this is now included.

Page 21-12, line 499-548: Details of how tumbling models are defined and constructed is now provided

Second, in our deeper investigation into the tumbling, we discovered a mistake in our code: the $T_{2,-1}$ and $T_{2,1}$ spherical tensors were swapped. This had no effect on rotating frame simulations, as must be the case, and also on lab frame simulations that did not involve multiple interactions. However, pseudocontact shift calculations and T_1 induced by CSA+dipole were affected, where rotations of interaction tensors were going in the incorrect direction for some terms. So, the reviewers will notice updates to Figure 3 F,H,I, and Figure 4B.

Thus far, this is not what the reviewer asked, but we thought we should first clarify due to changes in figures. The question is, are 10 sets good enough, and how do we get them? And so yes, 10 are good enough for the calculations shown, but in general the user needs to search for convergence with the number of Euler angles. In Methods, considerations for choosing tumbling models is not discussed.

Page 23, line 543-546: Discussion is added on care required when using tumbling models.

Page 10, line 221-222: We point the reader to the Supplementary information regarding proper tumbling model usage.

There is no discussion of powder averaging. On page 12, n_{gamma} is used but not described how to set it. In my opinion, insufficient references are used, for example the COMPUTE algorithm implemented here was described by Eden and Levitt before, γ -COMPUTE was introduced by Nielsen et al. ...

We have added an explanation of the powder averaging to the supplementary information

Page 11, line 248-250: We point out that the user needs to optimize powder settings

Page 15-17: We have added a section describing Euler angle definitions and powder averaging.

I think we didn't realize this was the COMPUTE algorithm. But indeed, it is at least something very similar. Levitt's theoretical work permeates the field rather thoroughly; occasionally I no longer realize that's where a mathematical concept I know actually comes from. Thanks for pointing this out to us.

Page 5, line 115: We cite the COMPUTE algorithm in reference to eigenbasis propagation

Page 19, line 448-449: We point out that our algorithm is closely related to the COMPUTE algorithm.

For calculations, all operators should be expressed in some basis. What is it for SLEEPY? Is it the eigenbasis of the Zeeman Hamiltonian or a basis of product operators? It becomes relevant when describing exchange operators L_{exc} as well as frequencies ω_{SR}

SLEEPY uses the basis set of the Zeeman Hamiltonian. This is a somewhat important point, because it more easily allows us to reduce the size of the Liouville matrix without approximating the spin-system behavior. (e.g. Kuprov uses $\hat{E}, \hat{S}^+, \hat{S}^-, \hat{S}_z$, which means each term has a well-defined spin-order, which they then approximate the system by limiting spin-order, allowing big reductions but yielding approximate solutions. We wanted exact simulations, but with some opportunity for simplification so we take the Zeeman basis to more easily allow exact basis set reductions).

Page 3, line 79-80: We briefly state what basis is used in SLEEPY

Page 20, line 474-479: The choice of basis set is discussed in more detail, with regards to basis set reduction.

Relaxation models are extremely difficult to set properly. When to use relaxation along cartesian coordinates and when in the eigenbasis? What type of thermalization to use? What is ρ_{eq} introduced in Eq.21? How to get it correctly? Is the approximation that all coherences decay with the same time constant always valid? And how to recognize numerical errors arising from the “dynamic thermalization” noted on page 20? As I said in the beginning, many difficult decisions are hidden...

These are all valid points. I think we will eventually expand the relaxation options in SLEEPY to include some means of implementing models yielding orientationally-dependent relaxation, and with some luck, stabilize the “dynamic thermalization” method (although this remains elusive). There isn’t really much choice in SLEEPY on the type of thermalization to use. If T_1 is introduced explicitly, then it automatically uses Lindblad thermalization, and if it arises from motion, then one must use “dynamic thermalization” (see discussion on page 27, line 631-633). In any case, we’ve added discussion about choosing the correct relaxation mode (eigenbasis or not), including that it may not always be appropriate to have the same T_1/T_2 for all orientations, and furthermore state that for “dynamic thermalization”, one should check that the system approaches the correct thermal equilibrium values.

Page 11-12, line 256-275: We have added discussion of the implemented methods for T_1 and T_2 and where to use them. We do state that a fixed relaxation rate constant may not be appropriate for all systems where multiexponential relaxation may occur.

Page 28, eq. 30: A definition for $\hat{\rho}_{eq}$ is added to the equation.

Page 29, line 660-661: We’ve pointed out that the user should check long-time evolution behavior to verify dynamic thermalization validity)

Minor discrepancies in Figure 1, (A) – $n=4000$ in the code but 6000 steps in the comment, (B) current time 88888,89 microsecond but 100 ms in the comment.

Page 6, Figure 1: Thanks for catching this. We have updated the figure accordingly.

Figure 2 – seqx seqmx in the legend, cwx and cwmx in the figures

Page 7, Figure 2: Thanks again for reading so closely. We have corrected the legend.